# Neural Ensemble Search
# for Uncertainty Estimation and Dataset Shift

**Sheheryar Zaidi**[1][*]    **Arber Zela**[2][*]    **Thomas Elsken**[3,2]
**Chris Holmes**[1]    **Frank Hutter**[2,3]    **Yee Whye Teh**[1]

[1]University of Oxford, [2]University of Freiburg, [3]Bosch Center for Artificial Intelligence
`{szaidi, cholmes, y.w.teh}@stats.ox.ac.uk,`
`{zelaa, fh}@cs.uni-freiburg.de, thomas.elsken@de.bosch.com`

## Abstract

Ensembles of neural networks achieve superior performance compared to stand-alone networks in terms of accuracy, uncertainty calibration and robustness to dataset shift. *Deep ensembles*, a state-of-the-art method for uncertainty estimation, only ensemble random initializations of a *fixed* architecture. Instead, we propose two methods for automatically constructing ensembles with *varying* architectures, which implicitly trade-off individual architectures' strengths against the ensemble's diversity and exploit architectural variation as a source of diversity. On a variety of classification tasks and modern architecture search spaces, we show that the resulting ensembles outperform deep ensembles not only in terms of accuracy but also uncertainty calibration and robustness to dataset shift. Our further analysis and ablation studies provide evidence of higher ensemble diversity due to architectural variation, resulting in ensembles that can outperform deep ensembles, even when having weaker average base learners. To foster reproducibility, our code is available: `https://github.com/automl/nes`

## 1   Introduction

Some applications of deep learning rely only on point estimate predictions made by a neural network. However, many critical applications also require reliable predictive uncertainty estimates and robustness under the presence of dataset shift, that is, when the observed data distribution at deployment differs from the training data distribution. Examples include medical imaging [15] and self-driving cars [5]. Unfortunately, several studies have shown that neural networks are not always robust to dataset shift [46, 26], nor do they exhibit calibrated predictive uncertainty, resulting in incorrect predictions made with high confidence [21].

*Deep ensembles* [33] achieve state-of-the-art results for predictive uncertainty calibration and robustness to dataset shift. Notably, they have been shown to outperform various approximate Bayesian neural networks [33, 46, 22]. Deep ensembles are constructed by training a *fixed* architecture multiple times with different random initializations. Due to the multi-modal loss landscape [18, 54], *randomization* by different initializations induces diversity among the base learners to yield a model with better uncertainty estimates than any of the individual base learners (i.e. ensemble members).

Our work focuses on *automatically* selecting *varying* base learner architectures in the ensemble, exploiting architectural variation as a beneficial source of diversity missing in deep ensembles due to their *fixed* architecture. Such architecture selection during ensemble construction allows a more "ensemble-aware" choice of architectures and is based on data rather than manual biases. As

---

[*]Equal contribution.

35th Conference on Neural Information Processing Systems (NeurIPS 2021).

discussed in Section 2, while ensembles with varying architectures has already been explored in the literature, variation in architectures is typically limited to just varying depth and/or width, in contrast to more complex variations, such as changes in the topology of the connections and operations used, as considered in our work. More generally, automatic ensemble construction is well-explored in AutoML [17, 36, 45, 43]. Our work builds on this by demonstrating that, in the context of *uncertainty estimation*, automatically constructed ensembles with varying architectures outperform deep ensembles that use state-of-the-art, or even *optimal*, architectures (Figure 1). Studied under controlled settings, we assess the ensembles by various measures, including predictive performance, uncertainty estimation and calibration, base learner performance and two ensemble diversity metrics, showing that architectural variation is beneficial in ensembles.

Note that, *a priori*, it is not obvious how to find a set of diverse architectures that work well as an ensemble. On the one hand, optimizing the base learners' architectures in isolation may yield multiple base learners with similar architectures (like a deep ensemble). On the other hand, selecting the architectures randomly may yield numerous base learners with poor architectures harming the ensemble. Moreover, as in neural architecture search (NAS), we face the challenge of needing to traverse vast architectural search spaces. We address these challenges in the problem of *Neural Ensemble Search* (NES), an extension of NAS that aims to find a *set* of complementary architectures that together form a strong ensemble. In summary, our contributions are as follows:

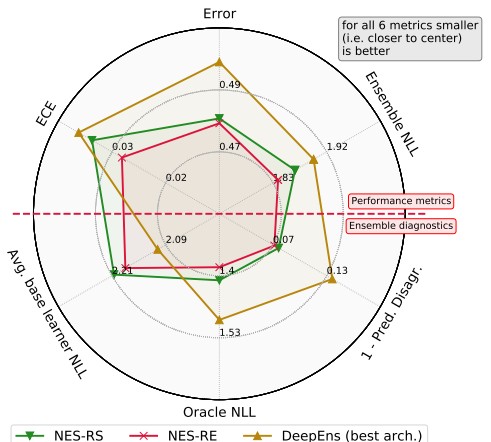

Figure 1: A comparison of a deep ensemble with the best architecture (out of 15,625 possible architectures) and ensembles constructed by our method NES on ImageNet-16-120 over NAS-Bench-201. Performance metrics (smaller is better): error, negative log-likelihood (NLL) and expected calibration error (ECE). We also measure average base learner NLL and two metrics for ensemble diversity (see Section 3): oracle NLL and [1−predictive disagreement]; small means more diversity for both metrics. NES ensembles outperform the deep ensemble, despite the latter having a significantly stronger average base learner.

1. We present two NES algorithms for automatically constructing ensembles with varying base learner architectures. As a first step, we present NES with random search (NES-RS), which is simple and easily parallelizable. We further propose NES-RE inspired by regularized evolution [48], which evolves a population of architectures yielding performant and robust ensembles.

2. This work is the first to apply automatic ensemble construction over architectures to *complex*, state-of-the-art neural architecture search spaces.

3. In the context of uncertainty estimation and robustness to dataset shift, we demonstrate that ensembles constructed by NES improve upon state-of-the-art deep ensembles. We validate our findings over five datasets and two architecture search spaces.

## 2 Related Work

**Ensembles and uncertainty estimation.** Ensembles of neural networks [23, 32, 11] are commonly used to boost performance. In practice, strategies for building ensembles include independently training multiple initializations of the same network, i.e. *deep ensembles* [33], training base learners on different bootstrap samples of the data [62], training with diversity-encouraging losses [40, 35, 60, 51, 29, 47] and using checkpoints during the training trajectory of a network [27, 41]. Despite a variety of approaches, Ashukha *et al.* [1] found many sophisticated ensembling techniques to be equivalent to a small-sized deep ensemble by test performance.

Much recent interest in ensembles has been due to their state-of-the-art predictive uncertainty estimates, with extensive empirical studies [46, 22] observing that deep ensembles outperform other approaches for uncertainty estimation, notably including Bayesian neural networks [4, 19, 52] and post-hoc calibration [21]. Although deep ensembles are not, technically speaking, equivalent to Bayesian neural networks and the relationship between the two is not well understood, diversity

among base learners in a deep ensemble yields a model which is arguably closer to *exact* Bayesian model averaging than other approximate Bayesian methods that only capture a single posterior mode in a multi-modal landscape [54, 18]. Also, He *et al.* [24] draw a rigorous link between Bayesian methods and deep ensembles for wide networks, and Pearce *et al.* [47] propose a technique for approximately Bayesian ensembling. Our primary baseline is deep ensembles as they provide state-of-the-art results in uncertainty estimation.

**AutoML and ensembles of varying architectures.** Automatic ensemble construction is commonly used in AutoML [17, 28]. Prior work includes use of Bayesian optimization to tune non-architectural hyperparameters of an ensemble's base learners [36], posthoc ensembling of fully-connected networks evaluated by Bayesian optimization [43] and building ensembles by iteratively adding (sub-)networks to improve ensemble performance [10, 42]. Various approaches, including ours, rely on ensemble selection [7]. We also note that Simonyan & Zisserman [49], He *et al.* [25] employ ensembles with varying architectures but *without* automatic construction. Importantly, in contrast to our work, all aforementioned works limit architectural variation to only changing width/depth or fully-connected networks. Moreover, such ensembles have not been considered before in terms of uncertainty estimation. Another important part of AutoML is neural architecture search (NAS), the process of automatically designing *single model* architectures [14], using strategies such as reinforcement learning [63], evolutionary algorithms [48] and gradient-based methods [39]. We use the search spaces defined by Liu *et al.* [39] and Dong & Yang [13], two of the most commonly used ones in recent literature.

Concurrent to our work, Wenzel *et al.* [53] consider ensembles with base learners having varying hyperparameters using an approach similar to NES-RS. However, they focus on non-architectural hyperparameters such as $L_2$ regularization strength and dropout rates, keeping the architecture fixed. As in our work, they also consider predictive uncertainty calibration and robustness to shift, finding similar improvements over deep ensembles.

## 3  Visualizing Ensembles of Varying Architectures

In this section, we discuss diversity in ensembles with varying architectures and visualize base learner predictions to add empirical evidence to the intuition that architectural variation results in more diversity. We also define two metrics for measuring diversity used later in Section 5.

### 3.1  Definitions and Set-up

Let $\mathcal{D}_{\text{train}} = \{(\boldsymbol{x}_i, y_i) : i = 1, \ldots, N\}$ be the training dataset, where the input $\boldsymbol{x}_i \in \mathbb{R}^D$ and, assuming a classification task, the output $y_i \in \{1, \ldots, C\}$. We use $\mathcal{D}_{\text{val}}$ and $\mathcal{D}_{\text{test}}$ for the validation and test datasets, respectively. Denote by $f_\theta$ a neural network with weights $\theta$, so $f_\theta(\boldsymbol{x}) \in \mathbb{R}^C$ is the predicted probability vector over the classes for input $\boldsymbol{x}$. Let $\ell(f_\theta(\boldsymbol{x}), y)$ be the neural network's loss for data point $(\boldsymbol{x}, y)$. Given $M$ networks $f_{\theta_1}, \ldots, f_{\theta_M}$, we construct the *ensemble $F$* of these networks by averaging the outputs, yielding $F(\boldsymbol{x}) = \frac{1}{M} \sum_{i=1}^{M} f_{\theta_i}(\boldsymbol{x})$.

In addition to the ensemble's loss $\ell(F(\boldsymbol{x}), y)$, we will also consider the *average base learner* loss and the *oracle ensemble's* loss. The average base learner loss is simply defined as $\frac{1}{M} \sum_{i=1}^{M} \ell(f_{\theta_i}(\boldsymbol{x}), y)$; we use this to measure the *average base learner strength* later. Similar to prior work [35, 60], the oracle ensemble $F_{\text{OE}}$ composed of base learners $f_{\theta_1}, \ldots, f_{\theta_M}$ is defined to be the function which, given an input $\boldsymbol{x}$, returns the prediction of the base learner with the smallest loss for $(\boldsymbol{x}, y)$, that is,

$$F_{\text{OE}}(\boldsymbol{x}) = f_{\theta_k}(\boldsymbol{x}), \quad \text{where} \quad k \in \underset{i}{\arg\min}\, \ell(f_{\theta_i}(\boldsymbol{x}), y).$$

The oracle ensemble can only be constructed if the true class $y$ is known. We use the oracle ensemble loss as one of the measures of *diversity* in base learner predictions. Intuitively, if base learners make diverse predictions for $\boldsymbol{x}$, the oracle ensemble is more likely to find some base learner with a small loss, whereas if all base learners make identical predictions, the oracle ensemble yields the same output as any (and all) base learners. Therefore, as a rule of thumb, all else being equal, smaller oracle ensemble loss indicates more diverse base learner predictions.

**Proposition 3.1.** Suppose $\ell$ is negative log-likelihood (NLL). Then, the oracle ensemble loss, ensemble loss, and average base learner loss satisfy the following inequality:

$$\ell(F_{\text{OE}}(\boldsymbol{x}), y) \leq \ell(F(\boldsymbol{x}), y) \leq \frac{1}{M} \sum_{i=1}^{M} \ell(f_{\theta_i}(\boldsymbol{x}), y).$$

We refer to Appendix A for a proof. Proposition 3.1 suggests that it can be beneficial for ensembles to not only have strong average base learners (smaller upper bound), but also more diversity in their predictions (smaller lower bound). There is extensive theoretical work relating strong base learner performance and diversity with the generalization properties of ensembles [23, 61, 6, 30, 3, 20]. In Section 5, the two metrics we use for measuring diversity are oracle ensemble loss and (normalized) predictive disagreement, defined as the average pairwise predictive disagreement amongst the base learners, normalized by their average error [18].

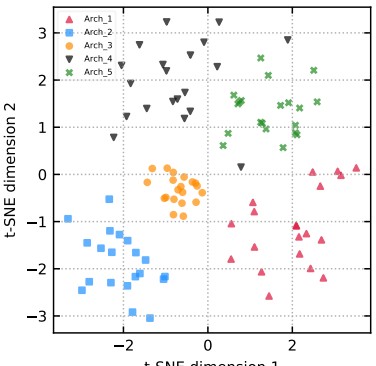

(a) Five different architectures, each trained with 20 different initializations.

### 3.2 Visualizing Similarity in Base Learner Predictions

The fixed architecture used to build deep ensembles is typically chosen to be a strong stand-alone architecture, either hand-crafted or found by NAS. However, optimizing the base learner's architecture and *then* constructing a deep ensemble can neglect diversity in favor of strong base learner performance. Having base learner architectures vary allows more diversity in their predictions. We provide empirical evidence for this intuition by visualizing the base learners' predictions. Fort *et al.* [18] found that base learners in a deep ensemble explore different parts of the function space by means of applying dimensionality reduction to their predictions. Building on this, we uniformly sample five architectures from the DARTS search space [39], train 20 initializations of each architecture on CIFAR-10 and visualize the similarity among the networks' predictions on the test dataset using t-SNE [50]. Experiment details are available in Section 5 and Appendix B.

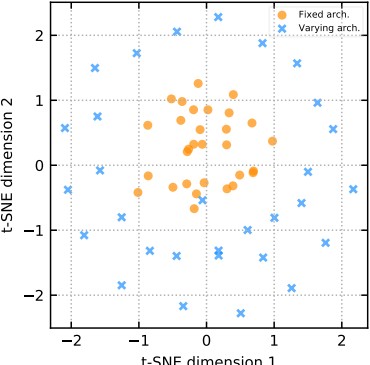

(b) Predictions of base learners in two ensembles, one with fixed architecture and one with varying architectures.

Figure 2: t-SNE visualization of base learner predictions. Each point corresponds to one network's (dimension-reduced) predictions.

As shown in Figure 2a, we observe clustering of predictions made by different initializations of a fixed architecture, suggesting that base learners with varying architectures explore different parts of the function space. Moreover, we also visualize the predictions of base learners of two ensembles, each of size $M = 30$, where one is a deep ensemble and the other has varying architectures (found by NES-RS as presented in Section 4). Figure 2b shows more diversity in the ensemble with varying architectures than in the deep ensemble. These qualitative findings can be quantified by measuring diversity: for the two ensembles shown in Figure 2b, we find the predictive disagreement to be 94.6% for the ensemble constructed by NES and 76.7% for the deep ensemble (this is consistent across independent runs). This indicates higher predictive diversity in the ensemble with varying architectures, in line with the t-SNE results.

## 4 Neural Ensemble Search

In this section, we define *neural ensemble search* (NES). In summary, a NES algorithm optimizes the architectures of base learners in an ensemble to minimize ensemble loss.

Given a network $f : \mathbb{R}^D \to \mathbb{R}^C$, let $\mathcal{L}(f, \mathcal{D}) = \sum_{(\boldsymbol{x}, y) \in \mathcal{D}} \ell(f(\boldsymbol{x}), y)$ be the loss of $f$ over dataset $\mathcal{D}$. Given a set of base learners $\{f_1, \ldots, f_M\}$, let Ensemble be the function which maps $\{f_1, \ldots, f_M\}$ to the ensemble $F = \frac{1}{M} \sum_{i=1}^{M} f_i$ as defined in Section 3 . To emphasize the architecture, we use the notation $f_{\theta, \alpha}$ to denote a network with architecture $\alpha \in \mathcal{A}$ and weights $\theta$, where $\mathcal{A}$ is a search space (SS) of architectures. A NES algorithm aims to solve the following optimization problem:

$$\min_{\alpha_1, \ldots, \alpha_M \in \mathcal{A}} \mathcal{L}\left(\text{Ensemble}(f_{\theta_1, \alpha_1}, \ldots, f_{\theta_M, \alpha_M}), \mathcal{D}_{\text{val}}\right) \tag{1}$$

$$\text{s.t.} \quad \theta_i \in \underset{\theta}{\arg\min} \, \mathcal{L}(f_{\theta, \alpha_i}, \mathcal{D}_{\text{train}}) \qquad \text{for } i = 1, \ldots, M$$

Eq. 1 is difficult to solve for at least two reasons. First, we are optimizing over $M$ architectures, so the search space is effectively $\mathcal{A}^M$, compared to it being $\mathcal{A}$ in typical NAS, making it more difficult

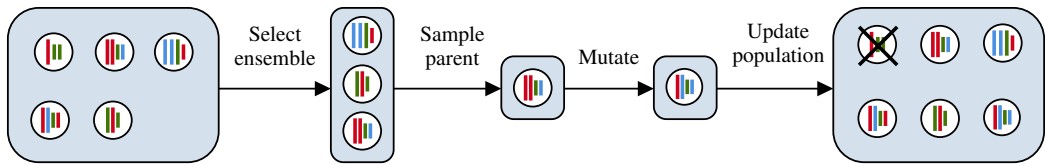

Figure 3: Illustration of one iteration of NES-RE. Network architectures are represented as colored bars of different lengths illustrating different layers and widths. Starting with the current population, ensemble selection is applied to select parent candidates, among which one is sampled as the parent. A mutated copy of the parent is added to the population, and the oldest member is removed.

to explore fully. Second, a larger search space also increases the risk of overfitting the ensemble loss to $\mathcal{D}_{\text{val}}$. A possible approach here is to consider the ensemble as a single large network to which we apply NAS, but joint training of an ensemble through a single loss has been empirically observed to underperform training base learners independently, specially for large networks [51]. Instead, our general approach to solve Eq. 1 consists of two steps:

1. **Pool building**: build a *pool* $\mathcal{P} = \{f_{\theta_1,\alpha_1}, \ldots, f_{\theta_K,\alpha_K}\}$ of size $K$ consisting of potential base learners, where each $f_{\theta_i,\alpha_i}$ is a network trained independently on $\mathcal{D}_{\text{train}}$.

2. **Ensemble selection**: select $M$ base learners (without replacement as discussed below) $f_{\theta_1^*,\alpha_1^*}, \ldots, f_{\theta_M^*,\alpha_M^*}$ from $\mathcal{P}$ to form an ensemble which minimizes loss on $\mathcal{D}_{\text{val}}$. (We set $K \geq M$.)

Step 1 reduces the options for the base learner architectures, with the intention to make the search more feasible and focus on strong architectures; step 2 then selects a performant ensemble. This procedure also ensures that the ensemble's base learners are trained independently. We use ensemble selection without replacement [7] for step 2. More specifically, this is forward step-wise selection; that is, given the set of networks $\mathcal{P}$, we start with an empty ensemble and add the network from $\mathcal{P}$ which minimizes ensemble loss on $\mathcal{D}_{\text{val}}$. We repeat this without replacement until the ensemble is of size $M$. `ForwardSelect`$(\mathcal{P}, \mathcal{D}_{\text{val}}, M)$ denotes the resulting set of $M$ base learners selected from $\mathcal{P}$.

Note that selecting the ensemble from $\mathcal{P}$ is a combinatorial optimization problem; a greedy approach such as `ForwardSelect` is nevertheless effective as shown by Caruana *et al.* [7], while keeping computational overhead low, given the predictions of the networks on $\mathcal{D}_{\text{val}}$. We also experimented with various other ensemble selection algorithms, including weighted averaging, as discussed in Section 5.2 and Appendix C.8, finding `ForwardSelect` to perform best.

We have not yet discussed the algorithm for building the pool in step 1; we present two approaches, NES-RS (Section 4.1) and NES-RE (Section 4.2). NES-RS is a simple random search based algorithm, while NES-RE is based on regularized evolution [48], a state-of-the-art NAS algorithm. Note that while gradient-based NAS methods have recently become popular, they are not naively applicable in our setting as the base learner selection component `ForwardSelect` is typically non-differentiable.

### 4.1 NES with Random Search

In NAS, random search (RS) is a competitive baseline on carefully designed architecture search spaces [37, 57, 58]. Motivated by its success and simplicity, NES with random search (NES-RS) builds the pool $\mathcal{P}$ by independently sampling architectures uniformly with replacement from the search space $\mathcal{A}$ (and training them). Since the architectures of networks in $\mathcal{P}$ vary, applying ensemble selection is a simple way to exploit diversity, yielding a performant ensemble. Importantly, NES-RS is easy to parallelize, exactly like deep ensembles. See Algorithm 2 in Appendix B.3 for pseudocode.

### 4.2 NES with Regularized Evolution

A more guided approach for building the pool $\mathcal{P}$ is using regularized evolution (RE) [48]. While RS has the benefit of simplicity by sampling architectures uniformly, the resulting pool might contain many weak architectures, leaving few strong architectures for `ForwardSelect` to choose between. RE is an evolutionary algorithm used on NAS spaces. It explores the search space by evolving (via mutations) a *population* of architectures. We first briefly describe RE as background before NES-RE. RE starts with a randomly initialized fixed-size population of architectures. At each iteration, a subset of size $m$ of the population is sampled, from which the best network by validation loss is selected as the parent. A mutated copy (e.g. changing an operation in the network, see Appendix

---

**Algorithm 1:** NES with Regularized Evolution

---

**Data:** Search space $\mathcal{A}$; ensemble size $M$; comp. budget $K$; $\mathcal{D}_{\text{train}}, \mathcal{D}_{\text{val}}$; population size $P$; number of parent candidates $m$.

**1** Sample $P$ architectures $\alpha_1, \ldots, \alpha_P$ independently and uniformly from $\mathcal{A}$.

**2** Train each architecture $\alpha_i$ using $\mathcal{D}_{\text{train}}$, and initialize $\mathfrak{p} = \mathcal{P} = \{f_{\theta_1, \alpha_1}, \ldots, f_{\theta_P, \alpha_P}\}$.

**3 while** $|\mathcal{P}| < K$ **do**

**4**     Select $m$ parent candidates $\{f_{\widetilde{\theta}_1, \widetilde{\alpha}_1}, \ldots, f_{\widetilde{\theta}_m, \widetilde{\alpha}_m}\} = \texttt{ForwardSelect}(\mathfrak{p}, \mathcal{D}_{\text{val}}, m)$.

**5**     Sample uniformly a parent architecture $\alpha$ from $\{\widetilde{\alpha}_1, \ldots, \widetilde{\alpha}_m\}$.     `// α stays in p.`

**6**     Apply mutation to $\alpha$, yielding child architecture $\beta$.

**7**     Train $\beta$ using $\mathcal{D}_{\text{train}}$ and add the trained network $f_{\theta, \beta}$ to $\mathfrak{p}$ and $\mathcal{P}$.

**8**     Remove the oldest member in $\mathfrak{p}$.           `// as done in RE [48].`

**9** Select base learners $\{f_{\theta_1^*, \alpha_1^*}, \ldots, f_{\theta_M^*, \alpha_M^*}\} = \texttt{ForwardSelect}(\mathcal{P}, \mathcal{D}_{\text{val}}, M)$ by forward step-wise selection without replacement.

**10 return** ensemble $\texttt{Ensemble}(f_{\theta_1^*, \alpha_1^*}, \ldots, f_{\theta_M^*, \alpha_M^*})$

---

B.4 for examples of mutations) of the parent architecture, called the child, is trained and added to the population, and the oldest member of the population is removed, preserving the population size. This is iterated until the computational budget is reached, returning the *history*, i.e. all the networks evaluated during the search, from which the best model is chosen by validation loss.

Building on RE for NAS, we propose NES-RE to build the pool of potential base learners. NES-RE starts by randomly initializing a population $\mathfrak{p}$ of size $P$. At each iteration, we first apply `ForwardSelect` to the population to select an ensemble of size $m$, then we uniformly sample one base learner from this ensemble to be the parent. A mutated copy of the parent is added to $\mathfrak{p}$ and the oldest network is removed, as in usual regularized evolution. This process is repeated until the computational budget is reached, and the history is returned as the pool $\mathcal{P}$. See Algorithm 1 for pseudocode and Figure 3 for an illustration.

Also, note the distinction between the *population* and the *pool* in NES-RE: the population is evolved, whereas the pool is the set of all networks evaluated during evolution (i.e., the history) and is used post-hoc for selecting the ensemble. Moreover, `ForwardSelect` is used both for selecting $m$ parent candidates (line 4 in NES-RE) and choosing the final ensemble of size $M$ (line 9 in NES-RE). In general, $m \neq M$.

### 4.3 Ensemble Adaptation to Dataset Shift

Deep ensembles offer (some) robustness to dataset shift relative to training data. In general, one may not know the type of shift that occurs at test time. By using an ensemble, diversity in base learner predictions prevents the model from relying on one base learner's predictions which may not only be incorrect but also overconfident.

We assume that one does not have access to data points with test-time shift at training time, but one does have access to some validation data $\mathcal{D}_{\text{val}}^{\text{shift}}$ with a *validation* shift, which encapsulates one's belief about test-time shift. Crucially, test and validation shifts are disjoint. To adapt NES-RS and NES-RE to return ensembles robust to shift, we propose using $\mathcal{D}_{\text{val}}^{\text{shift}}$ instead of $\mathcal{D}_{\text{val}}$ whenever applying `ForwardSelect` to select the final ensemble. In Algorithms 1 and 2, this is in lines 9 and 3, respectively. Our experiments show that this is highly effective against shift.

Note that in line 4 of Algorithm 1, we can also replace $\mathcal{D}_{\text{val}}$ with $\mathcal{D}_{\text{val}}^{\text{shift}}$ when expecting test-time shift; we simply sample one of $\mathcal{D}_{\text{val}}, \mathcal{D}_{\text{val}}^{\text{shift}}$ uniformly at each iteration, in order to promote exploration of architectures that work well both in-distribution and during shift and reduce cost by avoiding running NES-RE once for each of $\mathcal{D}_{\text{val}}, \mathcal{D}_{\text{val}}^{\text{shift}}$. See Appendices C.1.3 and B.4 for further discussion.

## 5 Experiments

### 5.1 Comparison to Baselines: Uncertainty Estimation & Robustness to Dataset Shift

We compare NES to deep ensembles on different choices of architecture search space (DARTS [39] and NAS-Bench-201 [13] search spaces) and dataset (Fashion-MNIST, CIFAR-10, CIFAR-100,

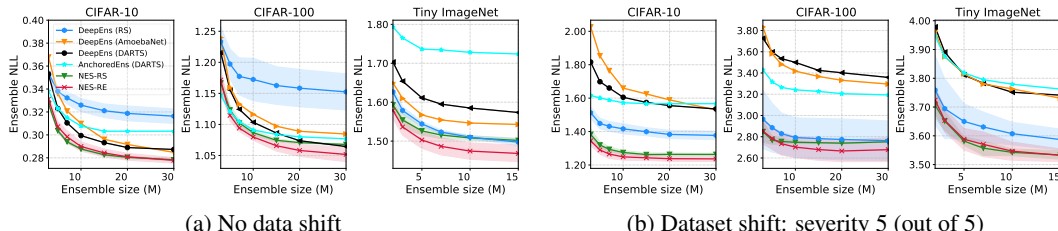

|  |  |
|---|---|
| (a) No data shift | (b) Dataset shift: severity 5 (out of 5) |

Figure 4: NLL vs. ensemble sizes on CIFAR-10, CIFAR-100 and Tiny ImageNet with and without dataset shifts [26] over the DARTS search space. Mean NLL shown with 95% confidence intervals.

ImageNet-16-120 and Tiny ImageNet). The search spaces are *cell-based*, containing a rich variety of convolutional architectures with differing cell topologies, number of connections and operations (see Appendix B.1 for visualizations). For CIFAR-10/100 and Tiny ImageNet, we also consider dataset shifts proposed by Hendrycks & Dietterich [26]. We use NLL, classification error and expected calibration error (ECE) [21, 44] as our metrics, for which small values are better. Note that NLL and ECE evaluate predictive uncertainty. Experimental/implementation details are in Appendix B, additional experiments are in Appendix C. All evaluations are on the test dataset. Each paragraph below highlights one of our main findings.

**Baselines.** One of our main baselines is deep ensembles with fixed, optimized architectures. We consider various optimized architectures, indicated as "DeepEns (`arch`)":

- On the DARTS search space, we consider the architectures found by:
  1. the DARTS algorithm (DeepEns (`DARTS`)),
  2. regularized evolution (DeepEns (`AmoebaNet`)),
  3. random search, *with the same number of networks trained* as NES algorithms (DeepEns (`RS`)).
- On the NAS-Bench-201 search space, in addition to DeepEns (`RS`), we consider:
  1. the architecture found by GDAS[2] [12] (DeepEns (`GDAS`)),
  2. the best architecture in the search space by validation loss (DeepEns (`best arch.`)).[3]

We also compare to *anchored ensembles* [47], a recent technique for approximately Bayesian ensembles, which explicitly regularizes each base learner towards a fresh initialization sample and aims to induce more ensemble diversity. We use the DARTS architecture, and our implementation is described in Appendix B.5.

**NES shows improved predictive uncertainty (NLL) and robustness to dataset shift (Figure 4).**
Figure 4a shows the NLL achieved by NES-RS, NES-RE and the baselines as functions of the ensemble size $M$ without dataset shift. We find that NES algorithms consistently outperform deep ensembles, with NES-RE usually outperforming NES-RS. Next, we evaluate the robustness of the ensembles to dataset shift in Figure 4b. In our setup, all base learners are trained on $\mathcal{D}_{\text{train}}$ without data augmentation of shifted examples. However, as explained in Section 4.3, we use a shifted validation dataset, $\mathcal{D}_{\text{val}}^{\text{shift}}$, and evaluate on a shifted test dataset, $\mathcal{D}_{\text{test}}^{\text{shift}}$. The types of shifts appearing in $\mathcal{D}_{\text{val}}^{\text{shift}}$ are disjoint from those in $\mathcal{D}_{\text{test}}^{\text{shift}}$. The severity of the shift varies between 1-5. We refer to Appendix B and Hendrycks & Dietterich [26] for details. The fixed architecture used in the baseline DeepEns (RS) is selected based on its loss over $\mathcal{D}_{\text{val}}^{\text{shift}}$, but the DARTS and AmoebaNet are architectures from the literature. As shown in Figure 4b, ensembles picked by NES-RS and NES-RE are significantly more robust to dataset shift than the baselines, highlighting the effectiveness of applying `ForwardSelect` with $\mathcal{D}_{\text{val}}^{\text{shift}}$. Unsurprisingly, AnchoredEns (DARTS) and DeepEns (DARTS/AmoebaNet) perform poorly compared to the other methods, as they are not optimized to deal with dataset shift here. The results of our experiments on Fashion-MNIST are in Appendix C.1.1. In line with the results in this section, both NES algorithms outperform deep ensembles with NES-RE performing best.

**Better uncertainty calibration versus dataset shift and classification error (Figure 5, Table 1).**
We also assess the ensembles by error and ECE. In short, ECE measures the mismatch between the model's confidence and accuracy. Figure 5 shows the ECE achieved by the ensembles at

---

[2]We did not consider the DARTS algorithm on NAS-Bench-201, since it returns degenerate architectures with poor performance on this space [13]. Whereas, GDAS yields state-of-the-art performance on this space.

[3]This is feasible, because all architectures in this search space were evaluated and are available.

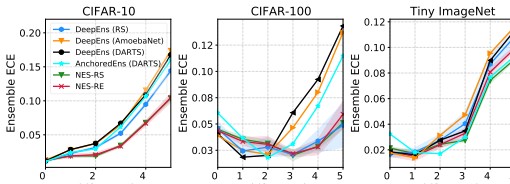
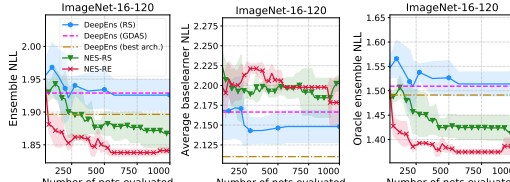

Figure 5: ECE vs. dataset shift severity on CIFAR-10, CIFAR-100 and Tiny ImageNet over the DARTS search space. No dataset shift is indicated as severity 0. Ensemble size is $M = 10$.

Figure 6: Ensemble, average base learner and oracle ensemble NLL versus budget $K$ on ImageNet-16-120 over the NAS-Bench-201 search space. Ensemble size is $M = 3$.

varying dataset shift severities, noting that uncertainty calibration is especially important when models are used during dataset shift. Ensembles found with NES tend to exhibit better uncertainty calibration and are either competitive with or outperform anchored and deep ensembles for most shift severities. Notably, on CIFAR-10, ECE is reduced by up to $40\%$ relative to the baselines. In terms of classification error, we find that ensembles constructed by NES outperform deep ensembles, with reductions of up to 7 percentage points in error, shown in Table 1. As with NLL, NES-RE tends to outperforms NES-RS.

**Ensembles found by NES tend to be more diverse (Table 3).** We measure ensemble diversity using the two metrics predictive disagreement (larger means more diverse) and oracle ensemble NLL (smaller means more diverse) as defined in Section 3 and average base learner NLL. Table 3 contrasts NES with the baselines in terms of these metrics. In terms of both diversity metrics, ensembles constructed by NES tend to be more diverse than anchored and deep ensembles. Ranking of the methods is largely consistent for different ensemble sizes $M$ (see Appendix C). Unsurprisingly, note that the average base learner in NES is not always the best: by optimizing the fixed architecture first and *then* ensembling it, deep ensembles end up with a strong average base learner at the expense of less ensemble diversity. Despite higher average base learner NLL, NES ensembles perform better (recall Figure 4), highlighting once again the importance of diversity.

**NES outperforms the deep ensemble of the best architecture in the NAS-Bench-201 search space (Figure 6).** Next, we compare NES to deep ensembles over the NAS-Bench-

t

Table 1: Classification error of ensembles for different shift severities. Best values and all values within $95\%$ confidence interval are bold faced. Note that NAS-Bench-201 comes with each architecture trained with *three* random initializations; therefore we set $M = 3$ in that case.

| Dataset | Shift Severity | Classif. error (%), $\mathcal{A}$ = DARTS search space | | | | | |
|---|---|---|---|---|---|---|---|
| | | DeepEns (RS) | DeepEns (Amoe.) | DeepEns (DARTS) | AnchoredEns (DARTS) | NES-RS | NES-RE |
| CIFAR-10 | 0 | $10.8_{\pm0.2}$ | 9.7 | 10.0 | 10.2 | $\mathbf{9.4_{\pm0.1}}$ | $\mathbf{9.4_{\pm0.2}}$ |
| | 3 | $25.1_{\pm0.7}$ | 25.6 | 26.3 | 28.6 | $23.2_{\pm0.2}$ | $\mathbf{22.9_{\pm0.2}}$ |
| | 5 | $41.1_{\pm0.9}$ | 42.7 | 42.9 | 44.9 | $38.0_{\pm0.2}$ | $\mathbf{37.4_{\pm0.4}}$ |
| CIFAR-100 | 0 | $33.2_{\pm1.2}$ | 31.6 | 30.9 | 30.9 | $30.7_{\pm0.1}$ | $\mathbf{30.4_{\pm0.4}}$ |
| | 3 | $54.8_{\pm1.6}$ | 54.2 | 55.1 | 55.2 | $\mathbf{49.8_{\pm0.1}}$ | $49.1_{\pm1.0}$ |
| | 5 | $64.3_{\pm3.2}$ | 68.5 | 68.5 | 68.9 | $62.4_{\pm0.2}$ | $\mathbf{61.4_{\pm1.4}}$ |
| Tiny ImageNet | 0 | $37.5_{\pm0.3}$ | 38.5 | 39.1 | 42.8 | $37.4_{\pm0.2}$ | $\mathbf{37.0_{\pm0.6}}$ |
| | 3 | $53.8_{\pm0.6}$ | 55.4 | 54.6 | 58.7 | $53.0_{\pm0.2}$ | $\mathbf{52.7_{\pm0.2}}$ |
| | 5 | $70.5_{\pm0.4}$ | 71.7 | 71.6 | 73.7 | $\mathbf{69.9_{\pm0.2}}$ | $70.2_{\pm0.1}$ |

(a) $M = 10$; DARTS search space.

| Dataset | Shift Severity | Classif. error (%), $\mathcal{A}$ = NAS-Bench-201 search space | | | | |
|---|---|---|---|---|---|---|
| | | DeepEns (GDAS) | DeepEns (best arch.) | DeepEns (RS) | NES-RS | NES-RE |
| CIFAR-10 | 0 | 8.4 | $\mathbf{7.2}$ | $7.8_{\pm0.2}$ | $7.7_{\pm0.1}$ | $7.6_{\pm0.1}$ |
| | 3 | 28.7 | 27.1 | $28.3_{\pm0.3}$ | $\mathbf{22.0_{\pm0.2}}$ | $22.5_{\pm0.1}$ |
| | 5 | 47.8 | 46.3 | $37.1_{\pm0.0}$ | $\mathbf{32.5_{\pm0.2}}$ | $33.0_{\pm0.5}$ |
| CIFAR-100 | 0 | 29.9 | 26.4 | $26.3_{\pm0.4}$ | $\mathbf{23.3_{\pm0.3}}$ | $\mathbf{23.5_{\pm0.2}}$ |
| | 3 | 60.3 | 54.5 | $57.0_{\pm0.9}$ | $\mathbf{46.6_{\pm0.3}}$ | $46.7_{\pm0.5}$ |
| | 5 | 75.3 | 69.9 | $64.5_{\pm0.0}$ | $\mathbf{59.7_{\pm0.2}}$ | $60.0_{\pm0.6}$ |
| ImageNet-16-120 | 0 | 49.9 | 49.9 | $50.5_{\pm0.6}$ | $48.1_{\pm1.0}$ | $\mathbf{47.9_{\pm0.4}}$ |

(b) $M = 3$; NAS-Bench-201 search space.

201 search space, which has two benefits: we demonstrate that our findings are not specific to the DARTS search space, and NAS-Bench-201 is an exhaustively evaluated search space for which all architectures' trained weights are available (three initializations per architecture), allowing us to compare NES to the deep ensemble of the *best* architecture by validation loss. Results shown in Figure 6 compare the losses of the ensemble, average base learner and oracle ensemble versus the number of networks evaluated $K$. Interestingly, although DeepEns (best arch.) has a significantly stronger average base learner than the other methods, its lack of diversity, as indicated by higher oracle ensemble loss (Figure 6) and lower predictive disagreement (Figure 1), yields a weaker ensemble than both NES algorithms. Also, NES-RE outperforms NES-RS with a 6.6x speedup as shown in Figure 6-left.

### 5.2 Analysis and Ablations

**Why does NES work? What if deep ensembles use ensemble selection over initializations? (Figure 7).** NES algorithms differ from deep ensembles in two important ways: the ensembles use

Table 3: Diversity and base learner strength. Predictive disagreement (larger means more diverse) and oracle ensemble NLL (smaller means more diverse) are defined in Section 3. Despite the stronger base learners, deep ensembles tend to be less diverse than ensembles constructed by NES. The results are consistent across datasets and shift severities (Appendix C). Best values and all values within 95% confidence interval are bold faced.

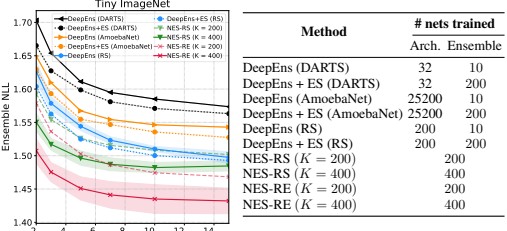

| Dataset | Metric | Method (with $M = 10$) | | | | | |
|---|---|---|---|---|---|---|---|
| | | DeepEns (RS) | DeepEns (Amoe.) | DeepEns (DARTS) | AnchoredEns (DARTS) | NES-RS | NES-RE |
| CIFAR-10 | Pred. Disagr. | $0.823_{\pm0.023}$ | **0.947** | 0.932 | 0.842 | $\mathbf{0.948}_{\pm0.004}$ | $0.943_{\pm0.009}$ |
| | Oracle NLL | $0.125_{\pm0.007}$ | 0.103 | 0.093 | 0.113 | $\mathbf{0.086}_{\pm0.001}$ | $0.088_{\pm0.002}$ |
| | Avg. bsl. NLL | $0.438_{\pm0.005}$ | 0.552 | 0.513 | **0.411** | $0.485_{\pm0.003}$ | $0.493_{\pm0.010}$ |
| CIFAR-100 | Pred. Disagr. | $0.831_{\pm0.027}$ | **0.946** | 0.935 | 0.839 | $0.934_{\pm0.006}$ | $0.943_{\pm0.014}$ |
| | Oracle NLL | $0.635_{\pm0.040}$ | 0.502 | 0.509 | 0.583 | $0.498_{\pm0.008}$ | $\mathbf{0.487}_{\pm0.014}$ |
| | Avg. bsl. NLL | $1.405_{\pm0.028}$ | 1.552 | 1.491 | **1.290** | $1.467_{\pm0.022}$ | $1.487_{\pm0.036}$ |
| Tiny ImageNet | Pred. Disagr. | $0.742_{\pm0.008}$ | 0.737 | 0.749 | 0.662 | $\mathbf{0.772}_{\pm0.005}$ | $0.768_{\pm0.005}$ |
| | Oracle NLL | $0.956_{\pm0.009}$ | 0.987 | 1.009 | 1.203 | $0.929_{\pm0.007}$ | $\mathbf{0.910}_{\pm0.015}$ |
| | Avg. bsl. NLL | $1.743_{\pm0.005}$ | 1.764 | 1.813 | 1.871 | $1.755_{\pm0.007}$ | $\mathbf{1.728}_{\pm0.012}$ |

Figure 7: Comparison of NES to deep ensembles with ensemble selection on Tiny ImageNet. LEFT: NLL vs ensemble size. RIGHT: Cost for Tiny ImageNet experiments reported in terms of the number of networks trained when $M = 10$. The "arch" column indicates the number of networks trained to first select an architecture, and the "ensemble" column contains the number of networks trained to build the ensemble.

varying architectures and NES utilizes ensemble selection (`ForwardSelect`) to pick the base learners. On Tiny ImageNet over the DARTS search space, we conduct an experiment to explore whether the improvement offered by NES over deep ensembles is only due to ensemble selection. The baselines "DeepEns + ES" operate as follows: we optimize a fixed architecture for the base learners, train $K$ random initializations of it to form a pool and apply `ForwardSelect` to select an ensemble of size $M$. Figure 7-left shows that both NES algorithms (each shown for two computational budgets $K = 200, 400$) outperform all DeepEns + ES baselines. Figure 7-right contains the cost of each method in terms of the number of networks trained. Note that DeepEns + ES (RS) is the most competitive of the deep ensemble baselines, and, at an equal budget of 400, it is outperformed by both NES algorithms. However, even at half the cost (200), NES-RE outperforms DeepEns + ES (RS) while NES-RS performs competitively. As expected, deep ensembles *with* ensemble selection consistently perform better than *without* ensemble selection at the expense of higher computational cost, but do not close the gap with NES algorithms. We also re-emphasize that comparisons between NES and deep ensembles in our experiments always fix the base learner training routine and method for composing the ensemble, so variations in ensemble performance are only due to the architecture choices. Therefore, to summarize, combined with the finding that NES outperforms deep ensembles even when NES' average base learner is weaker (as in e.g. Figure 6 and Table 3), we find architectural variation in ensembles to be important for NES' performance gains.

**Computational cost and parallelizability.** The primary computational cost involved in NES is training $K$ networks to build the pool $\mathcal{P}$. In our experiments on the DARTS search space, we set $K$ to be 400 for CIFAR-10/100 and 200 for Tiny ImageNet (except Figure 7 which additionally considers $K = 400$). Figure 7-right gives an example of costs for each method. The cost of a deep ensemble with an optimized architecture stems from the initial architecture search and the subsequent training of $M$ random initializations to build ensemble. We refer to Appendix B for details and discussion of computation cost, including training times. We note that apart from DeepEns (DARTS) and AnchoredEns (DARTS), NES has a lower computational cost than the baselines in our experiments. Similar to deep ensembles, training the pool for NES-RS is embarrassingly parallel. Our implementation of NES-RE is also parallelized as described in Appendix B.

**Comparison of different ensemble selection algorithms.** Both NES algorithms utilize `ForwardSelect` as the ensemble selection algorithm (ESA). We experimented with various other choices of ESAs, including weighted averaging and explicit diversity regularization, as shown in Figure 9. A detailed description of each ESA is in Appendix C.8. In summary, our choice of `ForwardSelect` performs better than or at par with all ESAs considered. Moreover, weighted averaging using stacking and Bayesian model averaging has

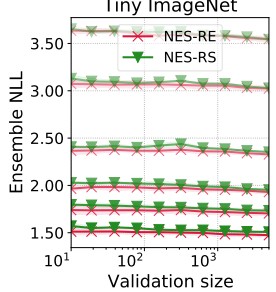

Figure 8: Test performance of NES algorithms with varying validation data sizes. Each curve corresponds to one particular dataset shift severity (0-5 with 0 being no shift). The more transparent curves correspond to higher shift severities.

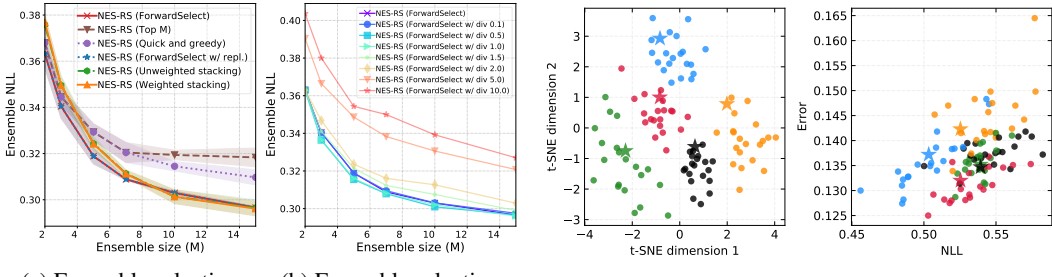

(a) Ensemble selection algorithms.

(b) Ensemble selection with explicit diversity regularization.

Figure 9: NES-RS on CIFAR-10.

Figure 10: LEFT: t-SNE of the test predictions of 20 single random mutations (circles) from 5 parent architecture (stars). RIGHT: NLL and error achieved by these architectures.

a very minor impact since the weights end up being close to uniform. Explicit diversity regularization, as shown in Figure 9b, appears to slightly improve performance in some cases, provided that the diversity regularization strength hyperparameter is appropriately tuned.

**NES is insensitive to the size of validation dataset** $\mathcal{D}_{\textbf{val}}$**.** We study the sensitivity of NES to the size of $\mathcal{D}_{\text{val}}$. Specifically, we measure test loss of the ensembles selected using $\mathcal{D}_{\text{val}}$ of different sizes (with as few as 10 validation samples). The results in Figure 8 indicate that NES is insensitive to validation size for different levels of dataset shift severity, achieving almost the same loss when using $10\times$ fewer validation data. We provide more details in Appendix C.5 and we also discuss why overfitting to $\mathcal{D}_{\text{val}}$ is averted during ensemble selection.

**Mutations of an architecture are similar in function space and by performance.** To explore how NES-RE traverses the search space, we analyzed how mutations affect an architecture as follows. We sampled five random architectures from the DARTS space, and for each one, we applied a single random mutation twenty times, yielding a "family" of parent-children architectures, which are then trained on CIFAR-10. Figure 10-left shows the result of t-SNE applied to the test predictions of these architectures, where each color corresponds to one "family", of which the "star" is the parent architecture and the circles are the children architectures. The clustering demonstrates that architectures which differ by only a single mutation are similar in function space after training. Figure 10-right shows the NLL and error achieved by these architectures. Again, similar clustering shows that architectures differing by a single mutation also perform similarly w.r.t. NLL and error. This confirms that mutations allow for locally exploring the search space.

**Further experiments.** We also provide additional experiments in the Appendix. This includes a comparison of ensembles built by averaging logits vs. probabilities (Appendix C.7), a comparison of NES to ensembles with other hyperparameters being varied (either width/depth or training hyperparameters similar to Wenzel *et al.* [53]) (Appendix C.4) and using a weight-sharing model [2] as a proxy to accelerate the search phase in NES (Appendix C.9).

## 6   Conclusion, Limitations & Broader Impact

We presented Neural Ensemble Search for automatically constructing ensembles with varying architectures and demonstrated that the resulting ensembles outperform state-of-the-art deep ensembles in terms of uncertainty estimation and robustness to dataset shift. Our work highlights the benefit of ensembling varying architectures. In future work, we aim to address the limitation posed by the computational cost of NES due to building the pool $\mathcal{P}$. An interesting approach in this direction could be the use of differentiable NAS methods to simultaneously optimize base learner architectures within a one-shot model and reduce cost [39, 56, 8]. More generally, we also hope to explore what other hyperparameters can be varied to improve ensemble performance and how best to select them.

Our work can readily be applied to many existing ensemble-based deep learning systems to improve their predictive performance. This work also focuses on improving uncertainty estimation in neural networks, which is a key problem of growing importance with implications for safe deployment of such systems. We are not aware of any direct negative societal impacts of our work, since NES is task-agnostic and its impact depends on its applications.

## Acknowledgments and Disclosure of Funding

AZ, TE and FH acknowledge support by the European Research Council (ERC) under the European Union Horizon 2020 research and innovation programme through grant no. 716721, and by BMBF grant DeToL. SZ acknowledges support from Aker Scholarship. CH wishes to acknowledge support from The Alan Turing Institute, The Medical Research Council UK, and the EPSRC Bayes4Health grant. The authors acknowledge support from ELLIS. We also thank Julien Siems for providing a parallel implementation of regularized evolution and Bobby He for useful comments on the manuscript.

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
