# Supplementary Material for Neural Ensemble Search for Uncertainty Estimation and Dataset Shift

## A Proof of Proposition 3.1

Taking the loss function to be NLL, we have $\ell(f(\boldsymbol{x}), y)) = -\log [f(\boldsymbol{x})]_y$, where $[f(\boldsymbol{x})]_y$ is the probability assigned by the network $f$ of $\boldsymbol{x}$ belonging to the true class $y$, i.e. indexing the predicted probabilities $f(\boldsymbol{x})$ with the true target $y$. Note that $t \mapsto -\log t$ is a convex and decreasing function.

We first prove $\ell(F_{\text{OE}}(\boldsymbol{x}), y) \leq \ell(F(\boldsymbol{x}), y)$. Recall, by definition of $F_{\text{OE}}$, we have $F_{\text{OE}}(\boldsymbol{x}) = f_{\theta_k}(\boldsymbol{x})$ where $k \in \arg\min_i \ell(f_{\theta_i}(\boldsymbol{x}), y)$, therefore $[F_{\text{OE}}(\boldsymbol{x})]_y = [f_{\theta_k}(\boldsymbol{x})]_y \geq [f_{\theta_i}(\boldsymbol{x})]_y$ for all $i = 1, \ldots, C$. That is, $f_{\theta_k}$ assigns the highest probability to the correct class $y$ for input $\boldsymbol{x}$. Since $-\log$ is a decreasing function, we have

$$\ell(F(\boldsymbol{x}), y) = -\log\left(\frac{1}{M}\sum_{i=1}^{M}[f_{\theta_i}(\boldsymbol{x})]_y\right)$$
$$\geq -\log\left([f_{\theta_k}(\boldsymbol{x})]_y\right) = \ell(F_{\text{OE}}(\boldsymbol{x}), y).$$

We apply Jensen's inequality in its finite form for the second inequality. Jensen's inequality states that for a real-valued, convex function $\varphi$ with its domain being a subset of $\mathbb{R}$ and numbers $t_1, \ldots, t_n$ in its domain, $\varphi(\frac{1}{n}\sum_{i=1}^{n} t_i) \leq \frac{1}{n}\sum_{i=1}^{n}\varphi(t_i)$. Noting that $-\log$ is a convex function, $\ell(F(\boldsymbol{x}), y) \leq \frac{1}{M}\sum_{i=1}^{M}\ell(f_{\theta_i}(\boldsymbol{x}), y)$ follows directly.

## B Experimental and Implementation Details

We describe details of the experiments shown in Section 5 and Appendix C and include Algorithms 2 and 3 describing NES-RS and `ForwardSelect`, respectively. Note that unless stated otherwise, all sampling over a discrete set is done uniformly in the discussion below.

---

**Algorithm 2:** NES with Random Search

    **Data:** Search space $\mathcal{A}$; ensemble size $M$; comp. budget $K$; $\mathcal{D}_{\text{train}}, \mathcal{D}_{\text{val}}$.
1  Sample $K$ architectures $\alpha_1, \ldots, \alpha_K$ independently and uniformly from $\mathcal{A}$.
2  Train each architecture $\alpha_i$ using $\mathcal{D}_{\text{train}}$, yielding a pool of networks
    $\mathcal{P} = \{f_{\theta_1, \alpha_1}, \ldots, f_{\theta_K, \alpha_K}\}$.
3  Select base learners $\{f_{\theta_1^*, \alpha_1^*}, \ldots, f_{\theta_M^*, \alpha_M^*}\} = \texttt{ForwardSelect}(\mathcal{P}, \mathcal{D}_{\text{val}}, M)$ by forward
    step-wise selection without replacement.
4  **return** ensemble $\texttt{Ensemble}(f_{\theta_1^*, \alpha_1^*}, \ldots, f_{\theta_M^*, \alpha_M^*})$

---

**Algorithm 3:** `ForwardSelect` (forward step-wise selection without replacement [7])

    **Data:** Pool of base learners $\mathcal{P}$; ensemble size $M$; $\mathcal{D}_{\text{val}}$. Assume $|\mathcal{P}| \geq M$.
1  Initialize an empty set of base learners $E = \{\}$.
2  **while** $|E| < M$ **do**
3     Add $f_{\theta, \alpha}$ to $E$, where $f_{\theta, \alpha} \in \arg\min_{f \in \mathcal{P}} \mathcal{L}(\texttt{Ensemble}(E \cup \{f\}), \mathcal{D}_{\text{val}})$
4     Remove $f_{\theta, \alpha}$ from $\mathcal{P}$.             // without replacement.
5  **return** $E$                     // set of selected base learners.

---

### B.1 Architecture Search Spaces

**DARTS search space.** The first architecture search space we consider in our experiments is the one from DARTS [39]. We search for two types of *cells*: *normal* cells, which preserve the spatial dimensions, and *reduction* cells, which reduce the spatial dimensions. See Figure 11b for an illustration. The cells are stacked in a macro architecture, where they are repeated and connected using skip connections (shown in Figure 11a). Each cell is a directed acyclic graph, where nodes represent feature maps in the computational graph and edges between them correspond to operation choices (e.g. a convolution operation). The cell parses inputs from the previous and previous-previous cells in its 2 input nodes. Afterwards it contains 5 nodes: 4 intermediate nodes that aggregate the

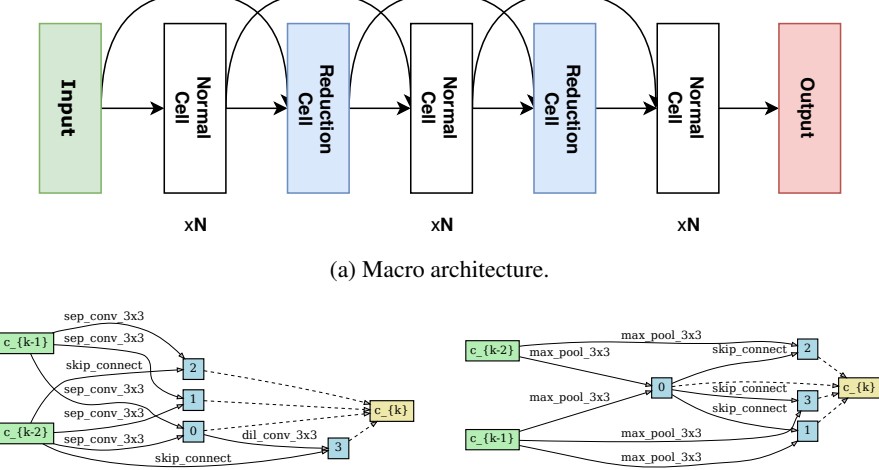

(a) Macro architecture.

(b) Normal (left) and reduction (right) cells.

Figure 11: Illustration of the DARTS search space: (a) The macro architecture is a stack of normal and reduction cells. Normal cells are repeated $N$ times between reduction cells ($N = 2$ in our experiments, i.e. 8 cells in total). (b) The cells for the DARTS architecture are depicted as directed acyclic graphs. Briefly, cells are composed of 2 input nodes (green), 4 intermediate nodes (blue) and one output node (red). Each edge is an operation which is applied to the preceding node's tensor and the output is summed element-wise in the succeeding node (an intermediate node). The output node is a concatenation of all intermediate nodes.

information coming from 2 previous nodes in the cell and finally an output node that concatenates the output of all intermediate nodes across the channel dimension. AmoebaNet contains one more intermediate node, making that a deeper architecture. The set of possible operations (eight in total in DARTS) that we use for each edge in the cells is the same as DARTS, but we leave out the "zero" operation since that is not necessary for non-differentiable approaches such as random search and evolution. Specifically, this leaves us with the following set of seven operations: $3 \times 3$ and $5 \times 5$ separable convolutions, $3 \times 3$ and $5 \times 5$ dilated separable convolutions, $3 \times 3$ average pooling, $3 \times 3$ max pooling and identity. Randomly of architectures is done by sampling the structure of the cell and the operations at each edge. The total number of architectures contained in this space is $\approx 10^{18}$. We refer the reader to Liu *et al.* [39] for more details.

**NAS-Bench-201 search space.** NAS-Bench-201 [13] is a tabular NAS benchmark, i.e. all architectures in the cell search space are trained and evaluated beforehand so one can query their performance (and weights) from a table quickly. Since this space is exhaustively evaluated, its size is also limited to only *normal* cells containing 4 nodes in total (1 input, 2 intermediate and 1 output node) and 5 operation choices on every edge connecting two nodes. This means that there are only 15,625 possible architecture configurations in this space. The networks are constructed by stacking 5 cells with in-between fixed residual blocks for reducing the spacial resolution. Each of them is trained for 200 epochs 3 times with 3 different seeds on 3 image classification datasets. For more details, please refer to Dong & Yang [13].

### B.2 Datasets

**Fashion-MNIST [55].** Fashion-MNIST consists of a training set of 60k $28 \times 28$ grayscale images and a test set of 10k images. The number of total labels is 10 classes. We split the 60k training set images to 50k used to train the networks and 10k used only for validation.

**CIFAR-10/100 [31].** CIFAR-10 and CIFAR-100 both consist of 60k $32 \times 32$ colour images with 10 and 100 classes, respectively. We use 10k of the 60k training images as the validation set. We use the 10k original test set for final evaluation.

**Tiny ImageNet [34].** Tiny Imagenet has 200 classes and each class has 500 training, 50 validation and 50 test colour images with $64 \times 64$ resolution. Since the original test labels are not available, we split the 10k validation examples into 5k for testing and 5k for validation.

**ImageNet-16-120 [13]**  This variant of the ImageNet-16-120 [9] contains 151.7k train, 3k validation and 3k test ImageNet images downsampled to 16×16 and 120 classes.

Note that the test data points are only used for final evaluation. The data points for validation are used by the NES algorithms and DeepEns + ES baselines during ensemble selection and by DeepEns (RS) for picking the best architecture from the pool to use in the deep ensemble. Note that when considering dataset shift for CIFAR-10, CIFAR-100 and Tiny ImageNet, we also apply two disjoint sets of "corruptions" (following the terminology used by Hendrycks & Dietterich [26]) to the validation and test sets. We never apply any corruption to the training data. More specifically, out of the 19 different corruptions provided by Hendrycks & Dietterich [26], we randomly apply one from {Speckle Noise, Gaussian Blur, Spatter, Saturate} to each data point in the validation set and one from {Gaussian Noise, Shot Noise, Impulse Noise, Defocus Blur, Glass Blur, Motion Blur, Zoom Blur, Snow, Frost, Fog, Brightness, Contrast, Elastic Transform, Pixelate, JPEG compression} to each data point in the test set. This choice of validation and test corruptions follows the recommendation of Hendrycks & Dietterich [26]. Also, as mentioned in Section 5, each of these corruptions has 5 severity levels, which yields 5 corresponding severity levels for $\mathcal{D}_{\text{val}}^{\text{shift}}$ and $\mathcal{D}_{\text{test}}^{\text{shift}}$.

### B.3  Training Routine & Time

The macro-architecture we use has 16 initial channels and 8 cells (6 normal and 2 reduction) and was trained using a batch size of 100 for 100 epochs for CIFAR-10/100 and 15 epochs for Fashion-MNIST. For Tiny ImageNet, we used 36 initial channels and a batch size of 128 for 100 epochs. Training a single network took roughly 40 minutes and 3 GPU hours[4] for CIFAR-10/100 and Tiny ImageNet, respectively. The networks are optimized using SGD with momentum set to 0.9. We used a learning rate of 0.025 for CIFAR-10/100 and Fashion-MNIST, and 0.1 for Tiny ImageNet. Unlike DARTS, we do not use any data augmentation procedure during training, nor any additional regularization such as ScheduledDropPath [64] or auxiliary heads, except for the case of Tiny ImageNet, for which we used ScheduledDropPath, gradient clipping and standard data augmentation as default in DARTS. All other hyperparameter settings are exactly as in DARTS [39].

All results containing error bars are averaged over multiple runs (at least five) with error bars indicating a 95% confidence interval. We used a budget $K = 400$ for CIFAR-10/100 (corresponding to 267 GPU hours). For Tiny ImageNet on the DARTS search space, unless otherwise stated, we used $K = 200$ (corresponding to 600 GPU hours). Note that only in Figure 7 and Table 30 we also tried NES algorithms using a higher budget of $K = 400$ on Tiny ImageNet for an equal-cost comparison with DeepEns + ES (RS). For ImageNet-16-120 on the NAS-Bench-201 search space, we used $K = 1000$ (compute costs are negligible since it is a tabular benchmark). Note that NES algorithms can be parallelized, especially NES-RS which is embarrassingly parallel like deep ensembles, therefore training time can be reduced easily when using multiple GPUs.

### B.4  Implementation Details of NES-RE

**Parallization.**  Running NES-RE on a single GPU requires evaluating hundreds of networks sequentially, which is tedious. To circumvent this, we distribute the "while $|\mathcal{P}| < K$" loop in Algorithm 1 over multiple GPUs, called worker nodes. We use the parallelism scheme provided by the `hpbandster` [16] codebase.[5] In brief, the master node keeps track of the population and history (lines 1, 4-6, 8 in Algorithm 1), and it distributes the training of the networks to the individual worker nodes (lines 2, 7 in Algorithm 1). In our experiments, we always use 20 worker nodes and evolve a population $\mathfrak{p}$ of size $P = 50$ when working over the DARTS search space. Over NAS-Bench-201, we used one worker since it is a tabular NAS benchmark and hence is quick to evaluate on. During iterations of evolution, we use an ensemble size of $m = 10$ to select parent candidates.

**Mutations.**  We adapt the mutations used in RE to the DARTS search space. As in RE, we first pick a normal or reduction cell at random to mutate and then sample one of the following mutations:

- `identity`: no mutation is applied to the cell.
- `op mutation`: sample one edge in the cell and replace its operation with another operation sampled from the list of operations described in Appendix B.1.

---

[4]We used NVIDIA RTX 2080Ti GPUs for training.
[5]https://github.com/automl/HpBandSter

- `hidden state mutation`: sample one intermediate node in the cell, then sample one of its two incoming edges. Replace the input node of that edge with another sampled node, without altering the edge's operation.

For example, one possible mutation would be changing the operation on an edge in the cell (as shown in e.g. Figure 16) from say $5 \times 5$ separable convolution to $3 \times 3$ max pooling. See Real *et al.* [48] for further details and illustrations of these mutations. Note that for NAS-Bench-201, following Dong & Yang [13] we only use `op mutation`.

**Adaptation of NES-RE to dataset shifts.** As described in Section 4.3, at each iteration of evolution, the validation set used in line 4 of Algorithm 1 is sampled uniformly between $\mathcal{D}_{\text{val}}$ and $\mathcal{D}_{\text{val}}^{\text{shift}}$ when dealing with dataset shift. In this case, we use shift severity level 5 for $\mathcal{D}_{\text{val}}^{\text{shift}}$. Once the evolution is complete and the pool $\mathcal{P}$ has been formed, then for each severity level $s \in \{0, 1, \ldots, 5\}$, we apply `ForwardSelect` with $\mathcal{D}_{\text{val}}^{\text{shift}}$ of severity $s$ to select an ensemble from $\mathcal{P}$ (line 9 in Algorithm 1), which is then evaluated on $\mathcal{D}_{\text{test}}^{\text{shift}}$ of severity $s$. (Here $s = 0$ corresponds to no shift.) This only applies to CIFAR-10, CIFAR-100 and Tiny ImageNet, as we do not consider dataset shift for Fashion-MNIST and ImageNet-16-120 .

### B.5 Implementation details of anchored ensembles.

Anchored ensembles [47] are constructed by independently training $M$ base learners, each regularized towards a new initialization sample (called the *anchor point*). Specifically, the $k$-th base learner $f_{\theta_k}$ is trained to minimize the following loss (using the notation in Section 3.1):

$$\frac{1}{N} \sum_{i=1}^{N} \ell(f_{\theta_k}(\boldsymbol{x}_i), y_i) + \frac{\lambda}{N} \|\Gamma^{1/2}(\theta_k - \mathring{\theta}_k)\|_2^2$$

where $\ell(f_{\theta_k}(\boldsymbol{x}_i), y_i) = -\log [f_{\theta_k}(\boldsymbol{x}_i)]_{y_i}$ is the cross-entropy loss for the $i$-th datapoint. Moreover, the anchor point $\mathring{\theta}_k$ is an independently sampled initialization, and $\lambda$ is the regularization strength. Defining $p$ to be the number of parameters, i.e. $\theta \in \mathbb{R}^p$, and letting $\sigma_i^2$ be the variance of the initialization for the $i$-th parameter $(\theta_k)_i$, $\Gamma \in \mathbb{R}^{p \times p}$ is the diagonal matrix with $\Gamma_{ii} = 1/(2\sigma_i^2)$.

Although Pearce *et al.* [47] do not include a tune-able regularization parameter $\lambda$ (they set $\lambda = 1$), we found anchored ensembles performed poorly without tuning $\lambda$. For CIFAR-10/100, we used $\lambda = 0.4$ and for Tiny ImageNet, we used $\lambda = 0.1$ (tuned by grid search). We followed the recommendation of Pearce *et al.* [47] in using a different initialization for optimization than the anchor point $\mathring{\theta}_k$. Moreover, we turned off weight decay when training anchored ensembles, and we did not apply the regularization to the trainable parameters in batch normalization layers (which are initialized deterministically).

## C  Additional Experiments

In this section we provide additional results for the experiments conducted in Section 5. Note that, as with all results shown in Section 5, all evaluations are made on test data unless stated otherwise.

### C.1  Supplementary plots complementing Section 5

### C.1.1  Results on Fashion-MNIST

As shown in Figure 12, we see a similar trend on Fashion-MNIST as with other datasets: NES ensembles outperform deep ensembles with NES-RE outperforming NES-RS. To understand why NES algorithms outperform deep ensembles on Fashion-MNIST [55], we compare the average base learner loss (Figure 13) and oracle ensemble loss (Figure 14) of NES-RS, NES-RE and DeepEns (RS). Notice that, apart from the case when ensemble size $M = 30$, NES-RS and NES-RE find ensembles with both stronger and more diverse base learners (smaller losses in Figures 13 and 14, respectively). While it is expected that the oracle ensemble loss is smaller for NES-RS and NES-RE compared to DeepEns (RS), it initially appears surprising that DeepEns (RS) has a larger average base learner loss considering that the architecture for the deep ensemble is chosen to minimize the base learner loss. We found that this is due to the loss having a sensitive dependence not only on the architecture but also the initialization of the base learner networks. Therefore, re-training the best architecture by validation loss to build the deep ensemble yields base learners with higher losses due to the

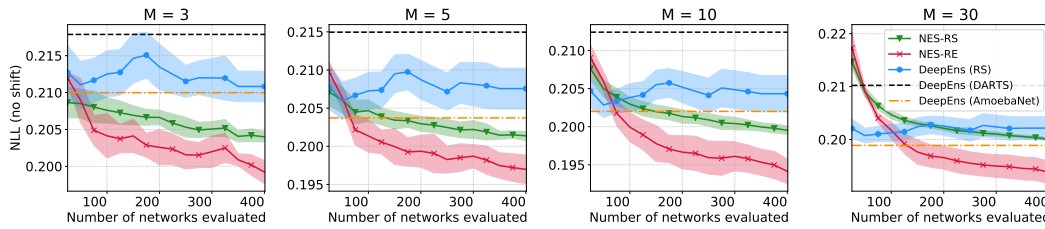

Figure 12: Results on Fashion-MNIST with varying ensembles sizes $M$. Lines show the mean NLL achieved by the ensembles with 95% confidence intervals.

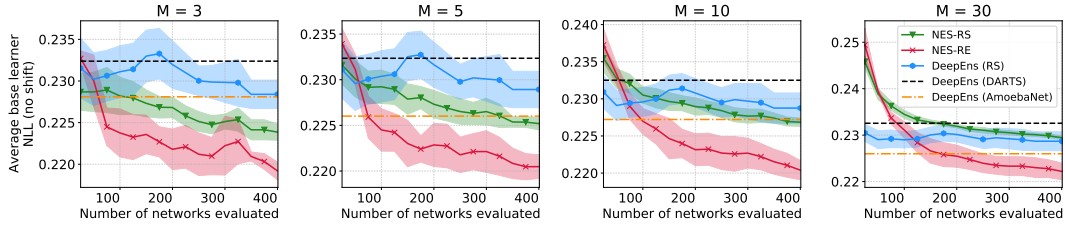

Figure 13: Average base learner loss for NES-RS, NES-RE and DeepEns (RS) on Fashion-MNIST. Lines show the mean NLL and 95% confidence intervals.

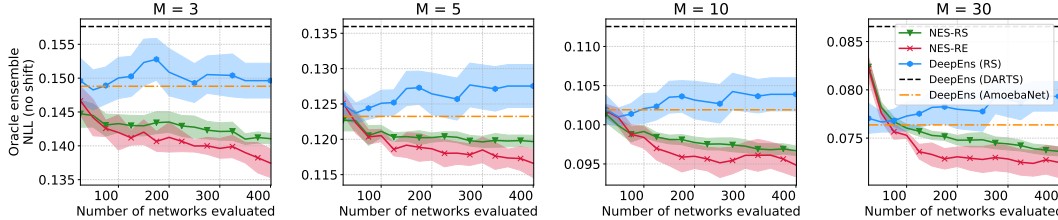

Figure 14: Oracle ensemble loss for NES-RS, NES-RE and DeepEns (RS) on Fashion-MNIST. Lines show the mean NLL and 95% confidence intervals.

use of different random initializations. Fortunately, NES algorithms are not affected by this, since they simply select the ensemble's base learners from the pool without having to re-train anything which allows them to exploit good architectures as well as initializations. Note that, for CIFAR-10-C experiments, this was not the case; base learner losses did not have as sensitive a dependence on the initialization as they did on the architecture.

In Table 4, we compare the classification error and expected calibration error (ECE) of NES algorithms with the deep ensembles baseline for various ensemble sizes on Fashion-MNIST. Similar to the loss, NES algorithms also achieve smaller errors, while ECE remains approximately the same for all methods.

### C.1.2 Entropy on out-of-distribution inputs

To assess how well models respond to completely out-of-distribution (OOD) inputs (inputs which do not belong to one of the classes the model can predict), we investigate the entropy of the predicted probability distribution over the classes when the input is OOD. Higher entropy of the predicted probabilities indicates more uncertainty in the model's output. For CIFAR-10 on the DARTS search space, we compare the entropy of the predictions made by NES ensembles with deep ensembles on two types of OOD inputs: images from the SVHN dataset and Gaussian noise. In Figure 15, we notice that NES ensembles indicate higher uncertainty when given inputs of Gaussian noise than deep ensembles but behave similarly to deep ensembles for inputs from SVHN.

### C.1.3 Additional Results on the DARTS search space

In this section, we provide additional experimental results on CIFAR-10, CIFAR-100 and Tiny ImageNet on the DARTS search space, complementing the results in Section 5 as shown in Figures 17-23. We also include examples of architectures chosen by NES-RE in Figure 16 for Tiny ImageNet, showcasing variation in architectures.

Table 4: Error and ECE of ensembles on Fashion-MNIST for different ensemble sizes $M$. Best values and all values within $95\%$ confidence interval are bold faced.

| $M$ | Classification Error (out of 1) | | | | | Expected Calibration Error (ECE) | | | | |
|---|---|---|---|---|---|---|---|---|---|---|
| | NES-RS | NES-RE | DeepEns (RS) | DeepEns (DARTS) | DeepEns (AmoebaNet) | NES-RS | NES-RE | DeepEns (RS) | DeepEns (DARTS) | DeepEns (AmoebaNet) |
| 3 | $0.074_{\pm 0.001}$ | $\mathbf{0.072}_{\pm 0.001}$ | $0.076_{\pm 0.001}$ | 0.077 | 0.077 | $0.007_{\pm 0.001}$ | $0.007_{\pm 0.002}$ | $0.008_{\pm 0.001}$ | $\mathbf{0.003}$ | 0.008 |
| 5 | $\mathbf{0.073}_{\pm 0.001}$ | $\mathbf{0.071}_{\pm 0.002}$ | $0.075_{\pm 0.001}$ | 0.077 | 0.074 | $\mathbf{0.005}_{\pm 0.001}$ | $\mathbf{0.005}_{\pm 0.001}$ | $\mathbf{0.006}_{\pm 0.001}$ | 0.005 | $\mathbf{0.005}$ |
| 10 | $0.073_{\pm 0.001}$ | $\mathbf{0.070}_{\pm 0.001}$ | $0.075_{\pm 0.001}$ | 0.076 | 0.073 | $\mathbf{0.004}_{\pm 0.001}$ | $0.005_{\pm 0.001}$ | $0.005_{\pm 0.001}$ | 0.006 | $\mathbf{0.005}$ |
| 30 | $0.073_{\pm 0.001}$ | $\mathbf{0.070}_{\pm 0.001}$ | $0.074_{\pm 0.001}$ | 0.075 | 0.073 | $\mathbf{0.004}_{\pm 0.001}$ | $\mathbf{0.004}_{\pm 0.002}$ | $\mathbf{0.004}_{\pm 0.001}$ | 0.008 | $\mathbf{0.004}$ |

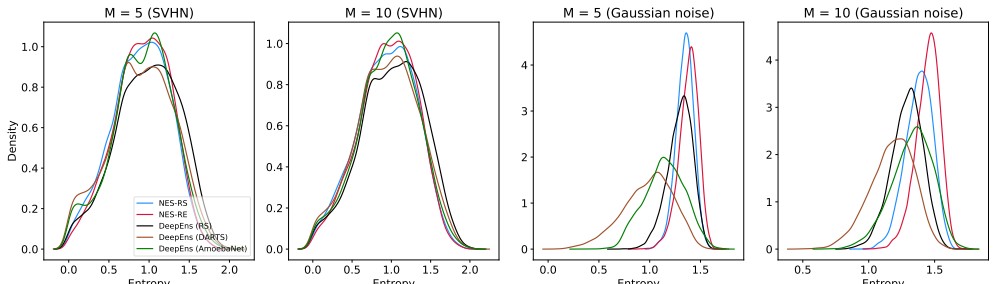

Figure 15: Entropy of predicted probabilities when trained on CIFAR-10 in the DARTS search space.

**Results on CIFAR for larger models.** In additional to the results on CIFAR-10 and CIFAR-100 on the DARTS search space using the settings described in Appendix B.3, we also train larger models (around 3M parameters) by scaling up the number of stacks cells and initial channels in the network. We run NES and other baselines similarly as done before and plot results in Figure 27 and 28 for NLL and classification test error with budget $K = 90$. As shown, NES algorithms tend to outperform or be competitive with the baselines. Note, more runs are needed including error bars for conclusive results in this case.

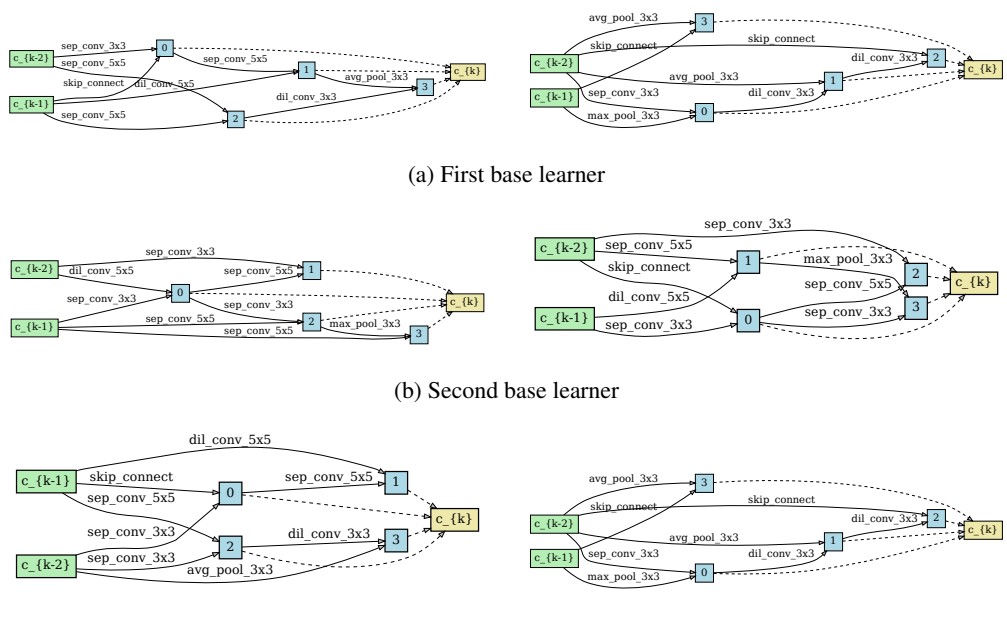

(a) First base learner

(b) Second base learner

(c) Third base learner

Figure 16: Illustration of example cells found by NES-RE for an ensemble of size $M = 3$ on Tiny ImageNet over the DARTS search space. In each sub-figure, left is the normal cell and right is the reduction cell. Note that the first and third base learners have the same reduction cell in this instance.

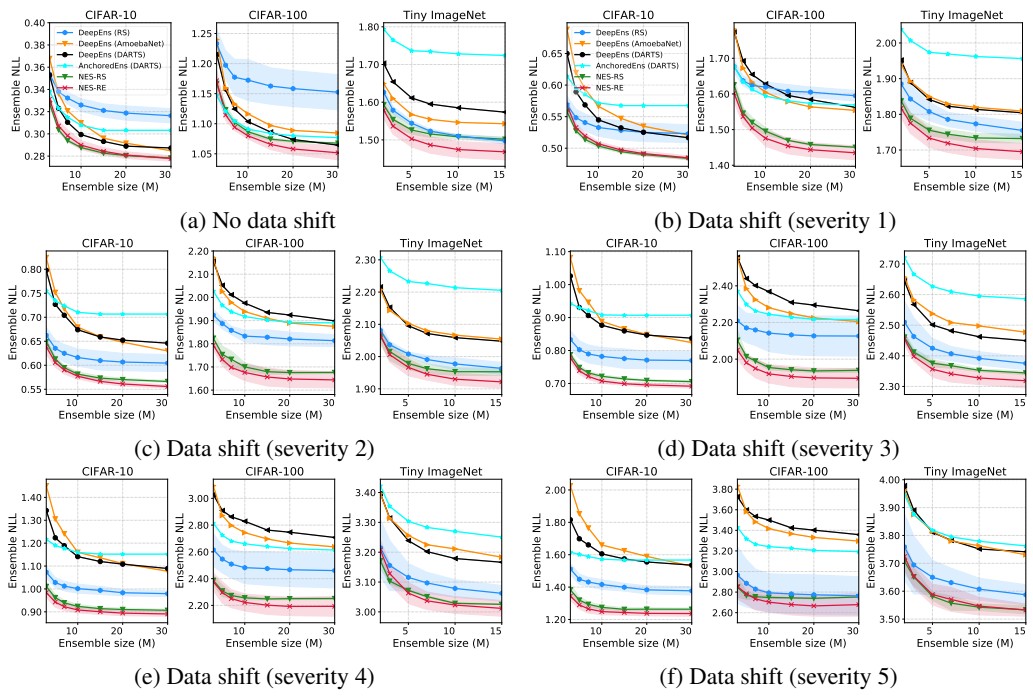

Figure 17: NLL vs. ensemble sizes on CIFAR-10, CIFAR-100 and Tiny ImageNet with varying dataset shifts [26] over DARTS space.

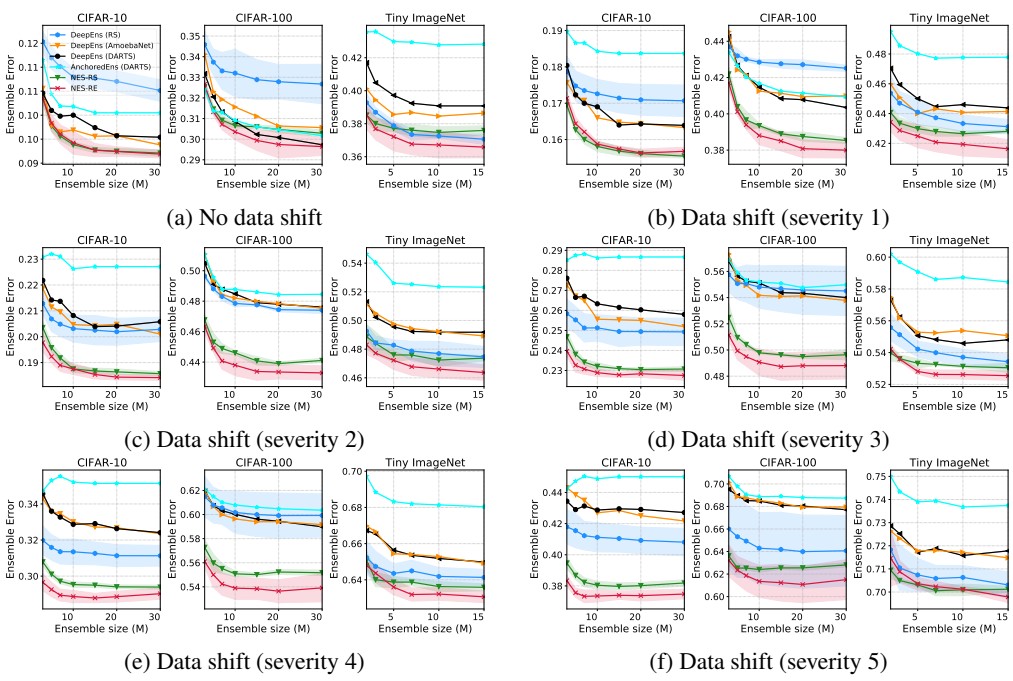

Figure 18: Classification error rate (between 0-1) vs. ensemble size on DARTS search space.

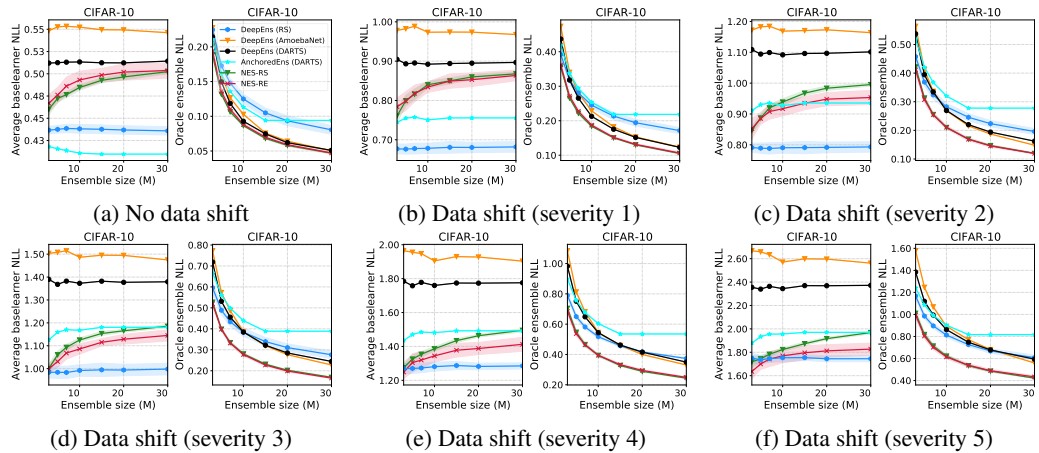

Figure 19: Average base learner and oracle ensemble NLL across ensemble sizes and shift severities on CIFAR-10 over DARTS search space.

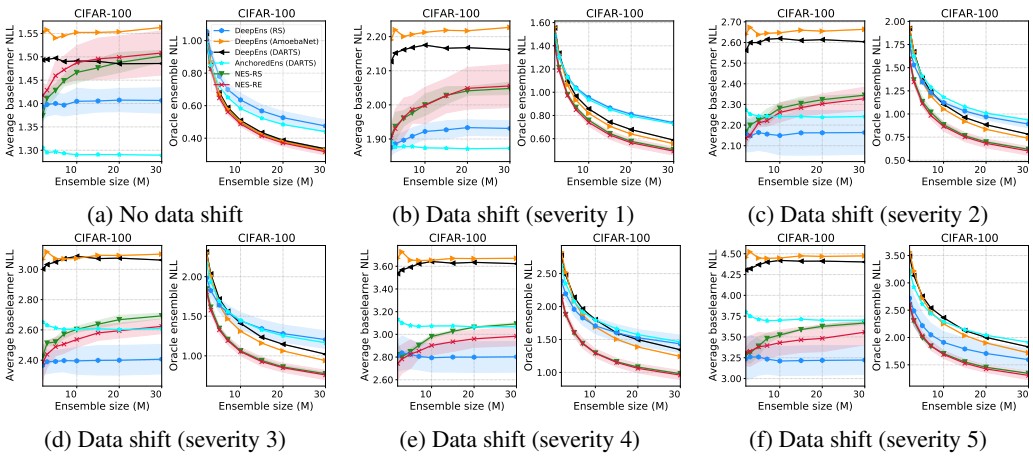

Figure 20: Average base learner and oracle ensemble NLL across ensemble sizes and shift severities on CIFAR-100 over DARTS search space.

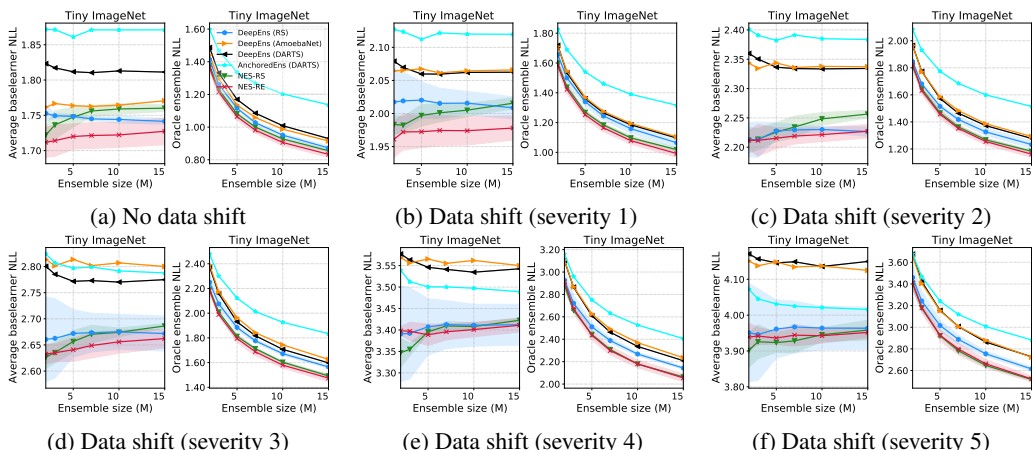

Figure 21: Average base learner and oracle ensemble NLL across ensemble sizes and shift severities on Tiny ImageNet over DARTS search space.

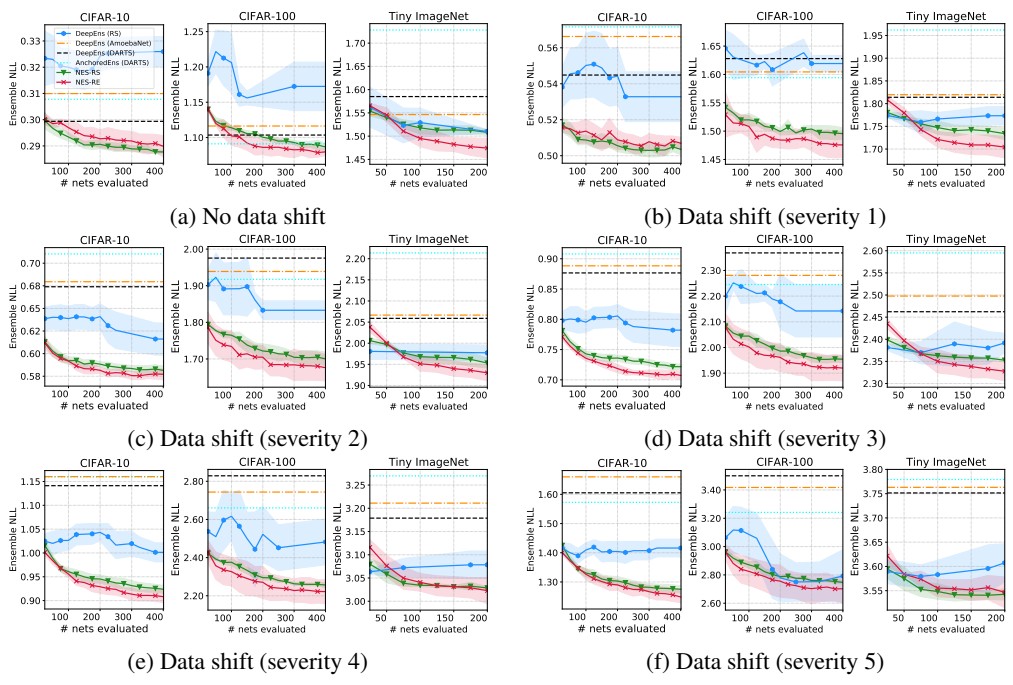

Figure 22: Ensemble NLL vs. budget $K$. Ensemble size fixed at $M = 10$.

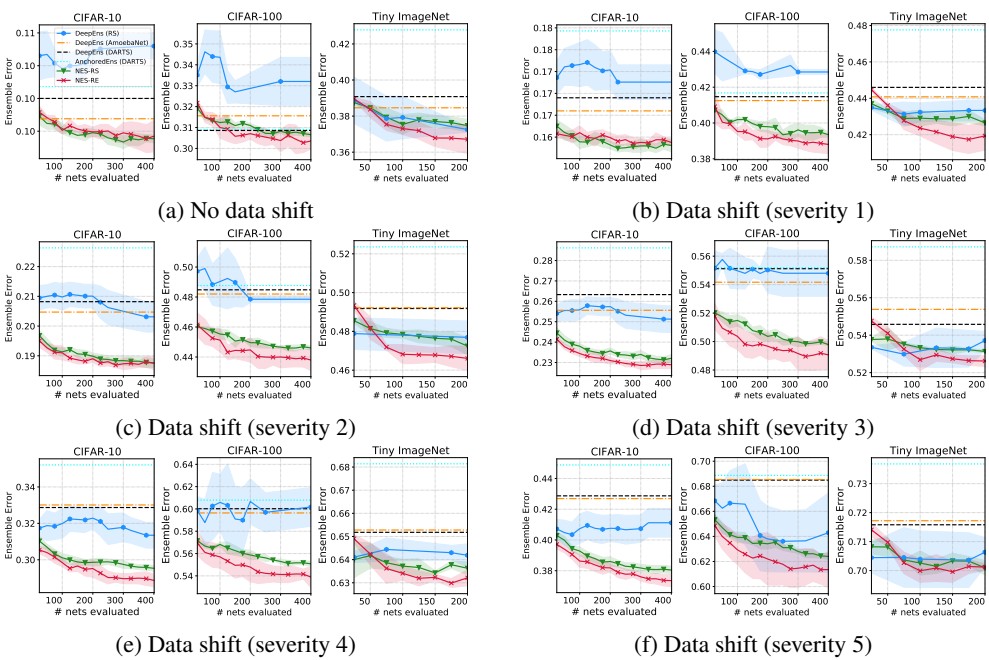

Figure 23: Ensemble error vs. budget $K$. Ensemble size fixed at $M = 10$.

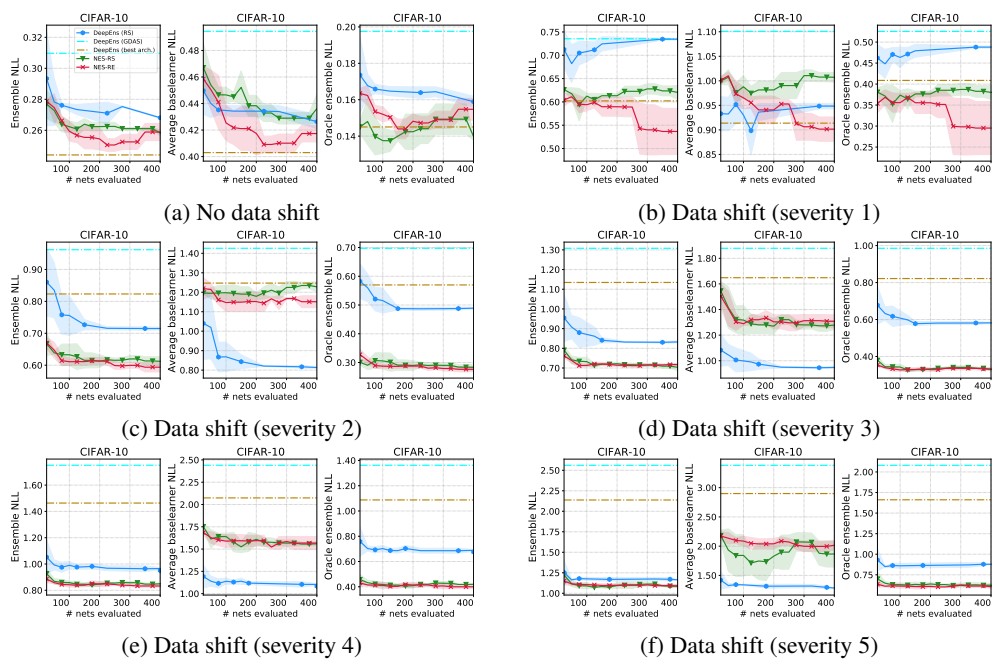

(a) No data shift  (b) Data shift (severity 1)

(c) Data shift (severity 2)  (d) Data shift (severity 3)

(e) Data shift (severity 4)  (f) Data shift (severity 5)

Figure 24: NES and deep ensembles performance on the NAS-Bench-201 search space and CIFAR-10 dataset. Plots show ensemble NLL, average baselearner NLL and oracle ensemble NLL vs. budget $K$. Ensemble size fixed at $M = 3$.

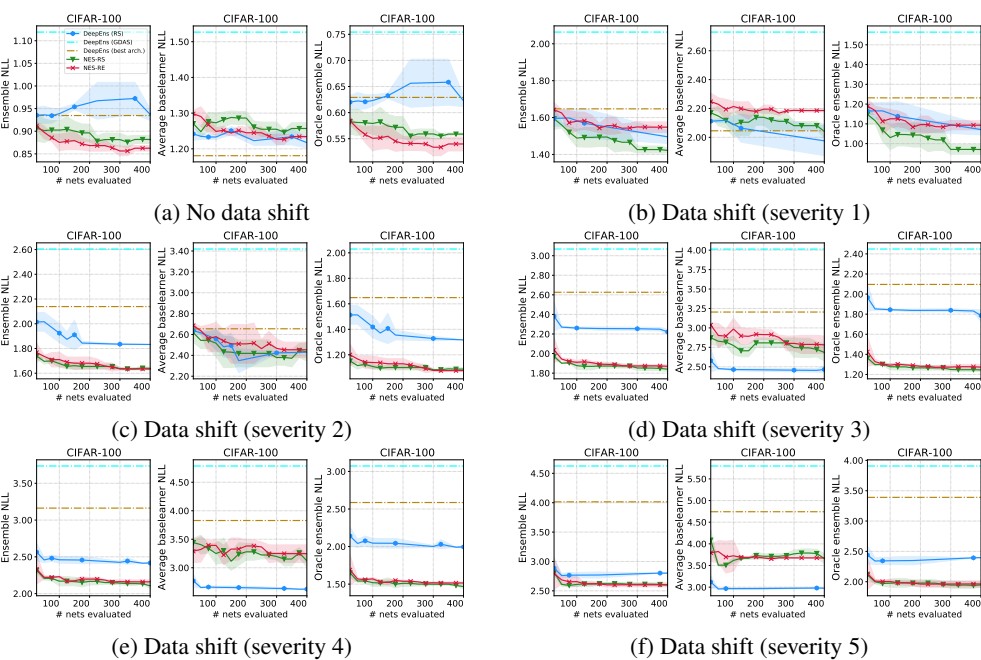

(a) No data shift  (b) Data shift (severity 1)

(c) Data shift (severity 2)  (d) Data shift (severity 3)

(e) Data shift (severity 4)  (f) Data shift (severity 5)

Figure 25: CIFAR-100 analogous to Figure 24.

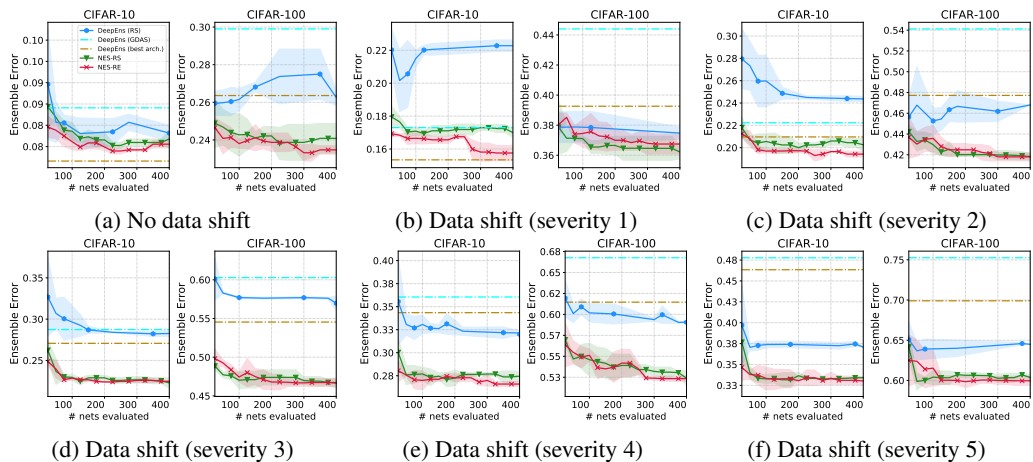

Figure 26: Ensemble error vs. budget $K$ on the NAS-Bench-201 space (CIFAR-10 and CIFAR-100). Ensemble size fixed at $M = 3$.

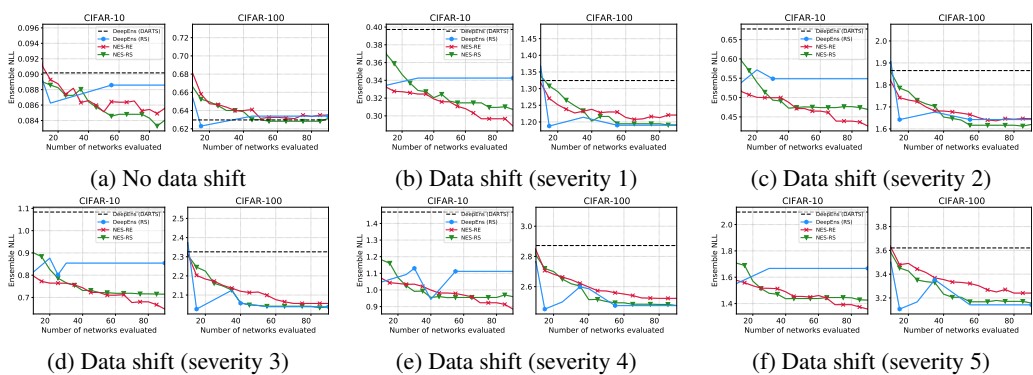

Figure 27: High fidelity NLL vs. budget $K$ on CIFAR-10 and CIFAR-100 with and without respective dataset shifts over the DARTS search space. Ensemble size is fixed at $M = 10$.

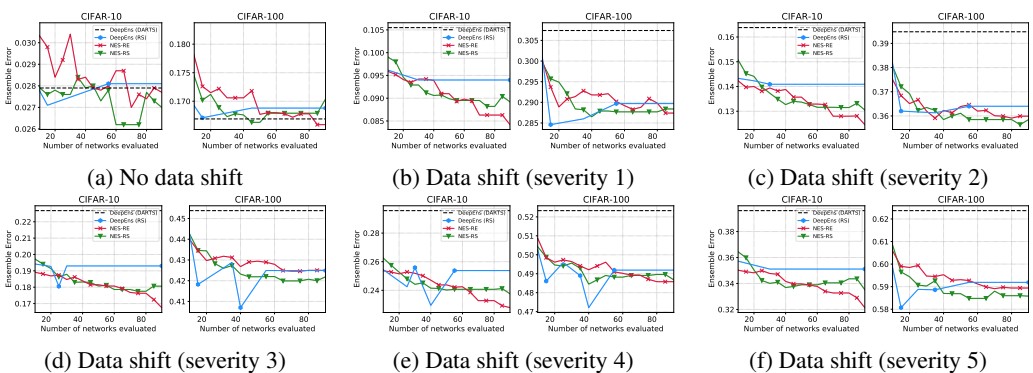

Figure 28: High fidelity classification error vs. budget $K$ on CIFAR-10 and CIFAR-100 with and without respective dataset shifts over the DARTS search space. Ensemble size is fixed at $M = 10$.

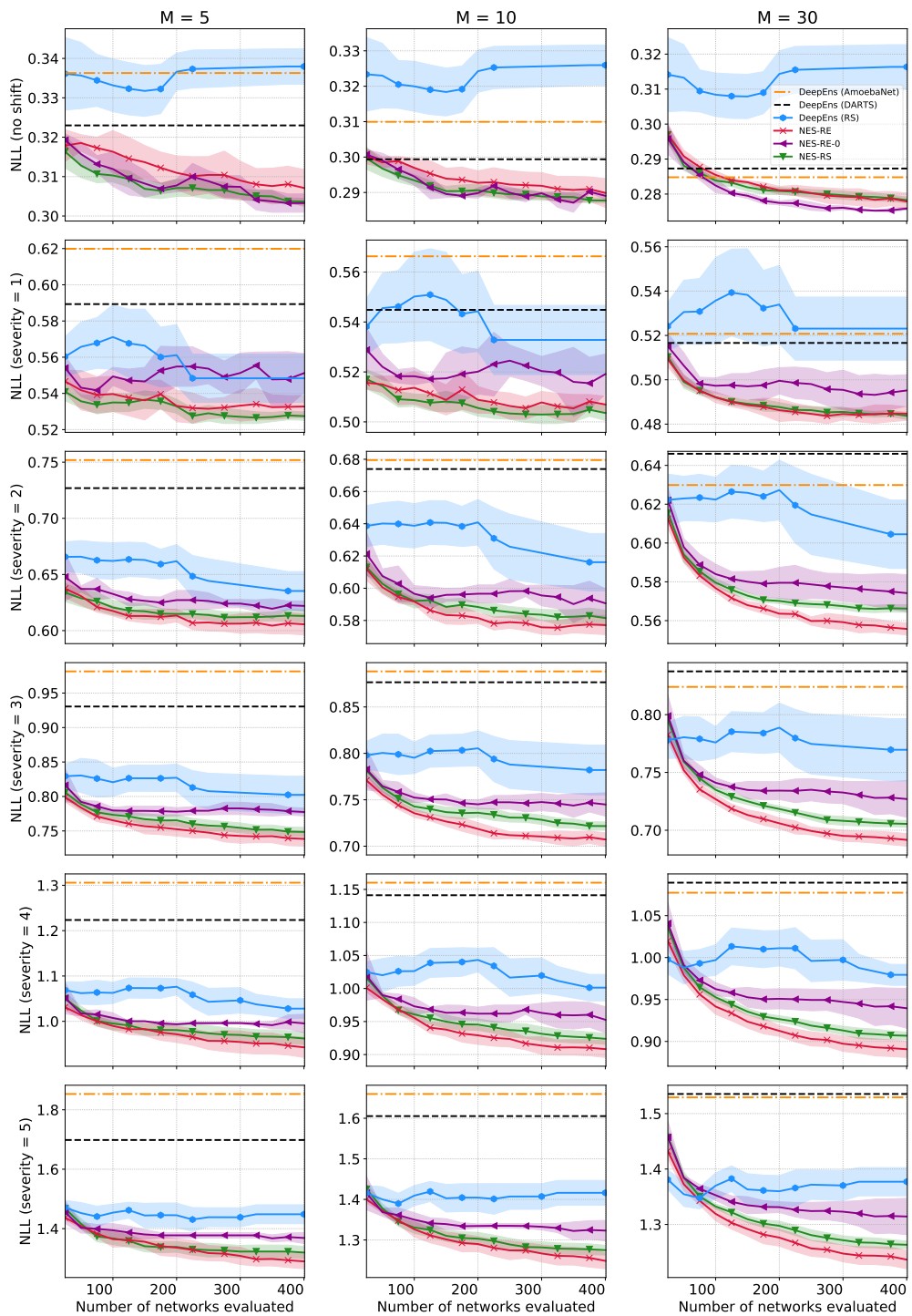

Figure 29: Results on CIFAR-10 [26] with varying ensembles sizes $M$ and shift severity. Lines show the mean NLL achieved by the ensembles with 95% confidence intervals. See Appendix C.1.3 for the definition of NES-RE-0.

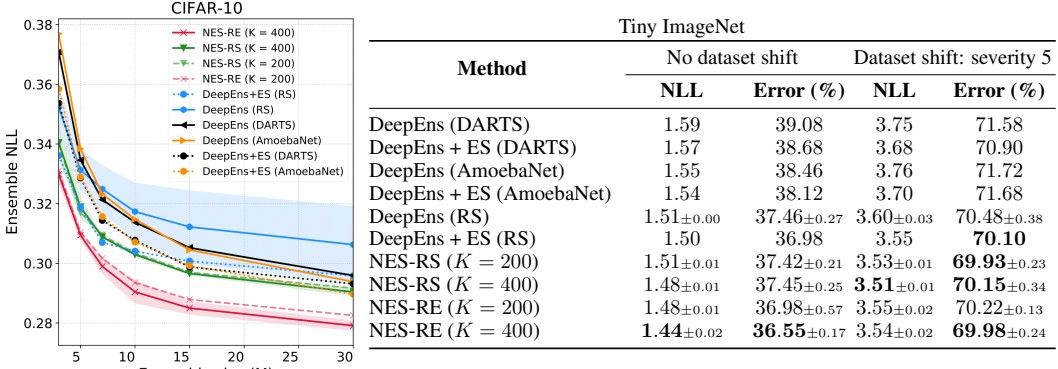

| Method | No dataset shift | | Dataset shift: severity 5 | |
| --- | --- | --- | --- | --- |
| | **NLL** | **Error (%)** | **NLL** | **Error (%)** |
| DeepEns (DARTS) | 1.59 | 39.08 | 3.75 | 71.58 |
| DeepEns + ES (DARTS) | 1.57 | 38.68 | 3.68 | 70.90 |
| DeepEns (AmoebaNet) | 1.55 | 38.46 | 3.76 | 71.72 |
| DeepEns + ES (AmoebaNet) | 1.54 | 38.12 | 3.70 | 71.68 |
| DeepEns (RS) | $1.51_{\pm0.00}$ | $37.46_{\pm0.27}$ | $3.60_{\pm0.03}$ | $70.48_{\pm0.38}$ |
| DeepEns + ES (RS) | 1.50 | 36.98 | 3.55 | **70.10** |
| NES-RS ($K=200$) | $1.51_{\pm0.01}$ | $37.42_{\pm0.21}$ | $3.53_{\pm0.01}$ | $\mathbf{69.93}_{\pm0.23}$ |
| NES-RS ($K=400$) | $1.48_{\pm0.01}$ | $37.45_{\pm0.25}$ | $\mathbf{3.51}_{\pm0.01}$ | $70.15_{\pm0.34}$ |
| NES-RE ($K=200$) | $1.48_{\pm0.01}$ | $36.98_{\pm0.57}$ | $3.55_{\pm0.02}$ | $70.22_{\pm0.13}$ |
| NES-RE ($K=400$) | $\mathbf{1.44}_{\pm0.02}$ | $\mathbf{36.55}_{\pm0.17}$ | $3.54_{\pm0.02}$ | $69.98_{\pm0.24}$ |

Figure 30: Comparison of NES to deep ensembles with ensemble selection over initializations over the DARTS search space. LEFT: NLL vs ensemble size on CIFAR-10 (similar to ablation in Figure 7). RIGHT: Test NLL and classification error NES vs. deep ensembles on Tiny ImageNet with ensemble size $M = 10$.

### C.1.4 Additional Results on the NAS-Bench-201 search space

In this section, we provide additional experimental results on CIFAR-10, CIFAR-100 and ImageNet-16-120 on the NAS-Bench-201 search space, complementing the results in Section 5 as shown in Figures 24-26.

### C.2 Ablation study: NES-RE optimizing only on $\mathcal{D}_{\text{val}}$

We also include a variant of NES-RE, called NES-RE-0, in Figure 29. NES-RE and NES-RE-0 are the same, except that NES-RE-0 uses the validation set $\mathcal{D}_{\text{val}}$ without any shift during iterations of evolution, as in line 4 of Algorithm 1. Following the discussion in Appendix B.4, recall that this is unlike NES-RE, where we sample the validation set to be either $\mathcal{D}_{\text{val}}$ or $\mathcal{D}_{\text{val}}^{\text{shift}}$ at each iteration of evolution. Therefore, NES-RE-0 evolves the population without taking into account dataset shift, with $\mathcal{D}_{\text{val}}^{\text{shift}}$ only being used for the post-hoc ensemble selection step in line 9 of Algorithm 1.

As shown in the Figure 29, NES-RE-0 shows a minor improvement over NES-RE in terms of loss for ensemble size $M = 30$ in the absence of dataset shift. This is in line with expectations, because evolution in NES-RE-0 focuses on finding base learners which form strong ensembles for in-distribution data. On the other hand, when there is dataset shift, the performance of NES-RE-0 ensembles degrades, yielding higher loss and error than both NES-RS and NES-RE. Nonetheless, NES-RE-0 still manages to outperform the DeepEns baselines consistently. We draw two conclusions on the basis of these results: (1) NES-RE-0 can be a competitive option in the absence of dataset shift. (2) Sampling the validation set, as done in NES-RE, to be $\mathcal{D}_{\text{val}}$ or $\mathcal{D}_{\text{val}}^{\text{shift}}$ in line 4 of Algorithm 1 plays an important role is returning a final pool $\mathcal{P}$ of base learners from which `ForwardSelect` can select ensembles robust to dataset shift.

### C.3 What if deep ensembles use ensemble selection over initializations?

We provide additional experimental results in this section for comparing NES to deep ensembles with ensemble selection building on those shown in Figure 7 in Section 5. The table in Figure 30-right compares the methods with respect to classification error as well as loss at different dataset shift severities. In the absence of any dataset shift, we find that NES-RE outperforms all baselines with respect to both metrics, specially at the higher budget of $K = 400$ and NES-RS performs competitively. At shift severity 5, NES algorithms also outperform the baselines, with NES-RS performing slightly better than NES-RE. The plot in Figure 30-left shows similar results on CIFAR-10 over the DARTS search space for the same experiment as the one conducted on Tiny ImageNet and shown in Figure 7.

### C.4 Comparing NES to ensembles with other varying hyperparameters

Since varying the architecture in an ensemble improves predictive performance and uncertainty estimation as demonstrated in Section 5, it is natural to ask what other hyperparameters should be

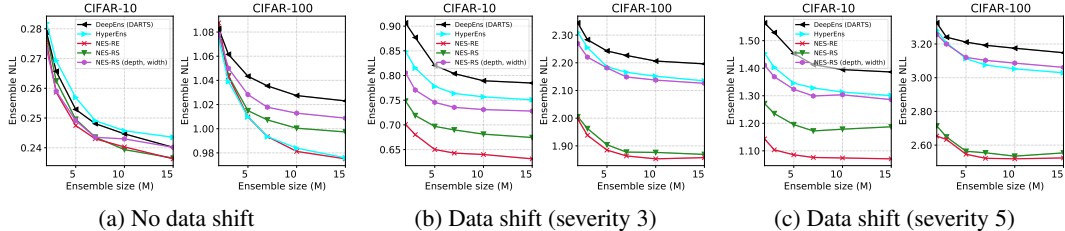

(a) No data shift     (b) Data shift (severity 3)     (c) Data shift (severity 5)

Figure 31: Plots show NLL vs. ensemble sizes comparing NES to the baselines introduced in Appendix C.4 on CIFAR-10 and CIFAR-100 with and without respective dataset shifts [26].

varied in an ensemble. It is also unclear which hyperparameters might be more important than others. Note that concurrent work by Wenzel *et al.* [53] has shown that varying hyperparameters such as $L_2$ regularization strength, dropout rate and label smoothing parameter also improves upon deep ensembles. While these questions lie outside the scope of our work and are left for future work, we conduct preliminary experiments to address them.

In this section, we consider two additional baselines working over the DARTS search space on CIFAR-10/100:

1. **HyperEns:** Optimize a fixed architecture, train $K$ random initializations of it *where the learning rate and $L_2$ regularization strength are also sampled randomly* and select the final ensemble of size $M$ from the pool using `ForwardSelect`. This is similar to `hyper ens` from Wenzel *et al.* [53].

2. **NES-RS (depth, width):** As described in Appendix B.1, NES navigates a complex (non-Euclidean) search space of architectures by varying the cell, which involves changing both the DAG structure of the cell and the operations at each edge of the DAG. We consider a baseline in which we keep the cell fixed (the optimized DARTS cell) and only vary the width and depth of the overall architecture. More specifically, we vary the number of *initial channels* $\in \{12, 14, 16, 18, 20\}$ (width) and the number of *layers* $\in \{5, 8, 11\}$ (depth). We apply NES-RS over this substantially simpler search space of architectures as usual: train $K$ randomly sampled architectures (i.e. sampling only depth and width) to form a pool and select the ensemble from it.

The results shown in Figures 31 and Table 32b compare the two baselines above to DeepEns (DARTS), NES-RS and NES-RE.[6] As shown in Figure 31, NES-RE tends to outperform the baselines, though is at par with HyperEns on CIFAR-100 without dataset shift (Figure 31a). Under the presence of dataset shift (Figures 31b and 31c), both NES algorithms substantially outperform all baselines. Note that both HyperEns and NES-RS (depth, width) follow the same protocol as NES-RS and NES-RE: ensemble selection uses a shifted validation dataset when evaluating on a

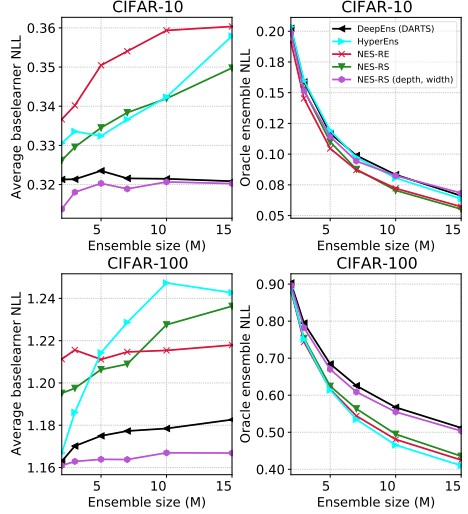

(a) Average base learner loss and oracle ensemble loss for NES and the baselines introduced in Appendix C.4 on CIFAR-10 and CIFAR-100. Recall that small oracle ensemble loss generally corresponds to higher diversity.

| Dataset | Shift Severity | DARTS search space | | | | |
|---|---|---|---|---|---|---|
| | | DeepEns (DARTS) | HyperEns | NES-RS (depth, width) | NES-RS | NES-RE |
| C10 | 0 | 8.2 | 8.1 | 8.0 | 8.0 | **7.7** |
| | 3 | 25.9 | 25.0 | 24.1 | 22.5 | **21.5** |
| | 5 | 43.3 | 40.8 | 42.1 | 38.1 | **34.9** |
| C100 | 0 | 28.8 | **28.1** | 28.8 | 28.4 | 28.4 |
| | 3 | 54.0 | 52.7 | 53.1 | 48.9 | **48.5** |
| | 5 | 68.4 | 67.2 | 67.9 | 61.3 | **60.7** |

(b) Classification errors comparing NES to the baselines introduced in Appendix C.4 for different shift severities and $M = 10$. Best values are bold faced.

Figure 32: See Appendix C.4 for details.

---

[6]Note that runs of DeepEns (DARTS), NES-RE and NES-RS differ slightly in this section relative to Section 5, as we tune the learning rate and $L_2$ regularization strength for each dataset instead of using the defaults used in Liu *et al.* [39]. This yields a fair comparison: HyperEns varies the learning rate and $L_2$ regularization while using a fixed, optimized architecture (DARTS), whereas NES varies the architecture while using fixed, optimized learning rate and $L_2$ regularization strength.

shifted test dataset. In terms of classification error, the observations are similar as shown in Table 32b. Lastly, we view the diversity of the ensembles from the perspective of oracle ensemble loss in Figure 32a. As in Section 5, results here also suggest that NES agorithms tend to find more diverse ensembles despite having higher average base learner loss.

## C.5 Sensitivity to the size of the validation dataset $\mathcal{D}_{\text{val}}$ and overfitting during ensemble selection

In this section, we consider how sensitive the performance of NES is to changes in the size of the validation dataset $\mathcal{D}_{\text{val}}$, and we discuss why overfitting to $\mathcal{D}_{\text{val}}$ is not a concern in our experiments. Recall that $\mathcal{D}_{\text{val}}$ is used by NES algorithms during ensemble selection from the pool of base learners as outlined in Algorithm 3, and we use $\mathcal{D}_{\text{val}}$ or $\mathcal{D}_{\text{val}}^{\text{shift}}$ depending on whether dataset shift is expected at test time as described in Section 4.3. Over the DARTS search space for CIFAR-100 and Tiny ImageNet, we measure the test loss of NES as a function of different validation dataset sizes. Specifically, we measure test loss of the ensembles selected using validation datasets of different sizes (with as few as 10 validation samples). Figure 33b shows this for different levels of dataset shift severity. The results indicate that NES is insensitive to the validation set size, achieving almost the same loss when using 10× fewer validation data.

NES also did not overfit to the validation data in our experiments. This can be seen in Figure 33a for CIFAR-100 over the DARTS search space, which shows that both the test and validation losses decrease over time (i.e. as the pool size $K$ increases). This finding is consistent across datasets and search spaces in our experiments. We also remark that overfitting is unexpected, because ensemble selection "fits a small number of parameters" since it only selects which base learners are added to the ensemble.

## C.6 Re-training the selected architectures on $\mathcal{D}_{\text{train}} + \mathcal{D}_{\text{val}}$

In practice, one might choose to re-trained the selected architectures from NES on the $\mathcal{D}_{\text{train}} + \mathcal{D}_{\text{val}}$ to make maximal use of the data available. We re-trained the ensembles constructed by NES and the best performing deep ensemble baseline, DeepEns (DARTS), on $\mathcal{D}_{\text{train}} + \mathcal{D}_{\text{val}}$, finding that both ensembles improve since the base learners improve due to more training data (see Figure 34). Note that, as shown in Appendix C.5, the performance of ensemble selection (which uses $\mathcal{D}_{\text{val}}$) is relatively insensitive to the size of $\mathcal{D}_{\text{val}}$. Therefore, one way to bypass the additional cost of having to re-train the ensembles on $\mathcal{D}_{\text{train}} + \mathcal{D}_{\text{val}}$ is to simply pick a very small $\mathcal{D}_{\text{val}}$, such that the performance of the models when trained on $\mathcal{D}_{\text{train}} + \mathcal{D}_{\text{val}}$ is approximately the same as when trained on $\mathcal{D}_{\text{train}}$.

## C.7 Averaging logits vs. averaging probabilities in an ensemble

In Figure 35, we explore whether ensembles should be constructed by averaging probabilities (i.e. post-softmax) as done in our work or by averaging the logits (i.e. pre-softmax). We find that while classification error of the resulting ensembles is similar, ensembles with averaged probabilities perform notably better in terms of uncertainty estimation (NLL) and are better calibrated as well (ECE).

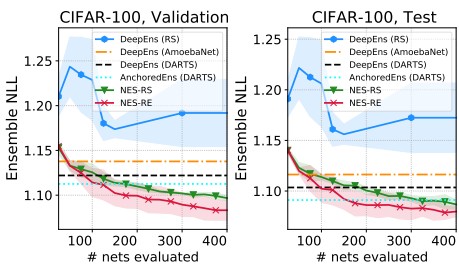

(a) Ensemble loss over the validation and test datasets over time (i.e. with increasing budget $K$). These curves being similar indicates no overfitting to the validation dataset during ensemble selection.

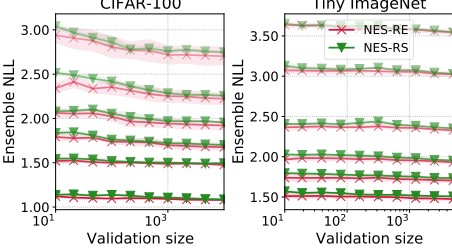

(b) Test performance of NES algorithms with varying validation data sizes. Each curve corresponds to one particular dataset shift severity (0-5 with 0 being no shift), with more transparent curves corresponding to higher shift severities. The loss stays relatively constant even with 10× fewer validation data samples, indicating NES is not very sensitive to validation dataset size.

Figure 33: See Appendix C.5 for details.

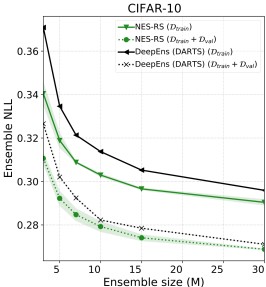

Figure 34: Re-training the selection architectures on $\mathcal{D}_{\text{train}} + \mathcal{D}_{\text{val}}$.

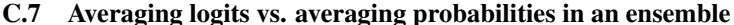

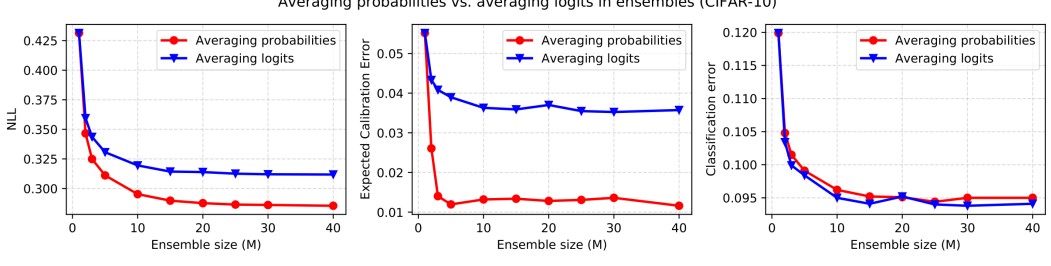

Figure 35: See Appendix C.7 for details.

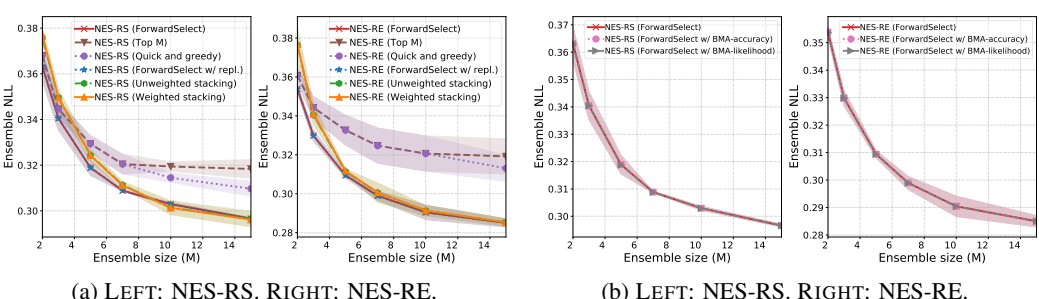

(a) LEFT: NES-RS. RIGHT: NES-RE.      (b) LEFT: NES-RS. RIGHT: NES-RE.

Figure 36: `ForwardSelect` compared to other ensemble selection algorithms. Experiments were conducted on CIFAR-10 over the DARTS search space. See Appendix C.8 for details.

## C.8   Comparison of ensemble selection algorithms

In Figure 9a, we compare the performance of NES for different choices of the ensemble selection algorithm (ESA) used in stage 2 (ensemble selection from pool). In addition to our default `ForwardSelect` without replacement, we experiment with the following seven choices:

- `TopM`: selects the top M (ensemble size) models by validation performance.
- `QuickAndGreedy`: Starting with the best network by validation performance, add the next best network to the ensemble only if it improves validation performance, iterating until the ensemble size is M or all models have been considered (returning an ensemble of size at most M).
- `ForwardSelect` with replacement: `ForwardSelect` but with replacement which allows repetitions of base learners.
- `Stacking` (Weighted): Linearly combines all base learners in the pool with learned stacking weights (which are positive and sum to 1). Then keeps only the base learners with the M largest stacking weights and weights them using the renormalized learned stacking weights.
- `Stacking` (Unweighted): Linearly combines all base learners in the pool with learned stacking weights (which are positive and sum to 1). Then keeps only the base learners with the M largest stacking weights and combines them by a simple, unweighted average.
- `ForwardSelect` with Bayesian model averaging by likelihood: Selects the base learners in the ensemble using `ForwardSelect` and then takes an average weighted by the (normalized) validation likelihoods of the base learners.
- `ForwardSelect` with Bayesian model averaging by accuracy: Selects the base learners in the ensemble using `ForwardSelect` and then takes an average weighted by the (normalized) validation accuracies of the base learners.

In Figure 36 we provide additional results including the one in Figure 9a. From the plots we can observe that:

- `ForwardSelect` performs better than or at par with all ESAs considered here.
- `Stacking` tends to perform competitively but still worse than `ForwardSelect`, and whether weighted averaging is used or not has a very minor impact on performance, since the learned weights tend to be close to uniform.

- Bayesian model averaging (BMA) also has a very minor impact (almost invisible in the plots in Figure 36b), because the likelihoods and accuracies of individual base learners are very similar (a consequence of multi-modal loss landscapes in neural networks with different models achieving similar losses) hence the BMA weights are close to uniform. The NLL achieved by the BMA ensemble only differed at the 4th or 5th decimal places compared to unweighted ensembles with the same base learners.

Lastly, we assess the impact of *explicitly* regularizing for diversity during ensemble selection as follows: we use `ForwardSelect` as before but instead use it to minimize the objective "validation loss - diversity strength × diversity" where diversity is defined as the average (across base learners and data points) $L_2$ distance between a base learner's predicted class probabilities and the ensemble's predicted class probabilities. The plots below show the results for different choices of the "diversity strength" hyperparameter. In summary, if appropriately tuned, `ForwardSelect` with diversity performs slightly better than usual `ForwardSelect` for NES-RS, though for NES-RE, the diversity term seems to harm performance as shown in Figure 37-right.

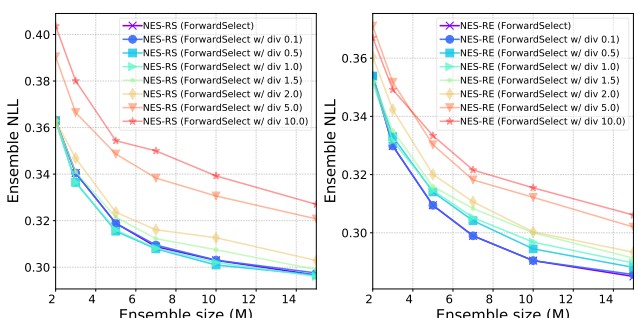

Figure 37: Ensemble selection with explicit diversity regularization. Experiments were conducted on CIFAR-10 over the DARTS search space.

## C.9  Generating the architecture pool using weight-sharing NAS algorithms

In order to accelerate the NES search phase, we generated the pool using the weight sharing schemes proposed by Random Search with Weight Sharing [37] and DARTS [39]. Specifically, we trained one-shot weight-sharing models using each of these two algorithms, then we sampled architectures from the weight-shared models uniformly at random to build the pool. Next, we ran `ForwardSelect` as usual to select ensembles from these pools. Finally, we retrained the selected base learners from scratch using the same training pipeline as NES and the deep ensemble baselines. In Figure 38, we refer to the results of these methods as NES-

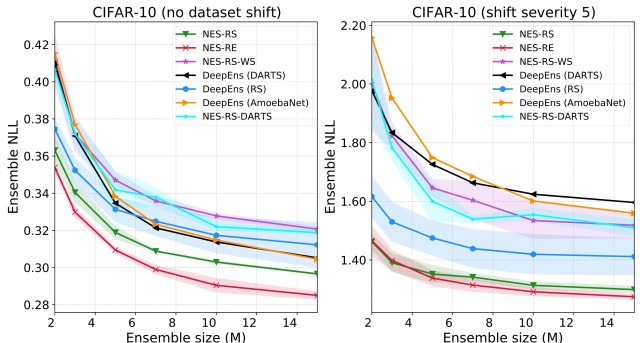

Figure 38: Comparison of NES-RS-WS and NES-RS-DARTS with the other baselines and NES-RS and NES-RE. The plots show the NLL vs. ensemble sizes on CIFAR-10 with and without dataset shifts over the DARTS search space. Mean NLL shown with 95 confidence intervals.

RS-WS and NES-RS-DARTS, respectively. As we can see, their performance is worse than NES-RS and NES-RE, even though the computation cost is reduced significantly. This is likely due to the low correlation between the performance of the architectures when evaluated using the shared weights and the performance when re-trained in isolation. Prior work has shown this is caused by weight interference and co-adaptation which occurs during the weight-sharing model training [58, 59]. Nonetheless, we believe this is a promising avenue for future research requiring more development, as it substantially reduces the cost of exploring the search space.

# D   NAS Best Practice Checklist

Our experimental setup resembles that of NAS papers, therefore, to foster reproducibility, we now describe how we addressed the individual points of the NAS best practice checklist by Lindauer & Hutter [38].

1. **Best Practices for Releasing Code**

   For all experiments you report:
   (a) Did you release code for the training pipeline used to evaluate the final architectures? [Yes] Code for our experiments, including NES and baselines, is open-source.
   (b) Did you release code for the search space? [Yes]
   (c) Did you release the hyperparameters used for the final evaluation pipeline, as well as random seeds? [Yes]
   (d) Did you release code for your NAS method? [Yes]
   (e) Did you release hyperparameters for your NAS method, as well as random seeds? [Yes]

2. **Best practices for comparing NAS methods**
   (a) For all NAS methods you compare, did you use exactly the same NAS benchmark, including the same dataset (with the same training-test split), search space and code for training the architectures and hyperparameters for that code? [Yes] All base learners follow the same training routine and use architectures from the same search space.
   (b) Did you control for confounding factors (different hardware, versions of DL libraries, different runtimes for the different methods)? [Yes]
   (c) Did you run ablation studies? [Yes] See Section 5.2 and Appendices C.2, C.3, C.4, C.5, C.6, C.7, C.8, C.9.
   (d) Did you use the same evaluation protocol for the methods being compared? [Yes]
   (e) Did you compare performance over time? [Yes] As shown in all figures with $K$ (which is proportional to training time) on the $x$-axis.
   (f) Did you compare to random search? [Yes]
   (g) Did you perform multiple runs of your experiments and report seeds? [Yes] We ran NES with multiple seeds and averaged over them in Section 5.
   (h) Did you use tabular or surrogate benchmarks for in-depth evaluations? [Yes] We performed experiments over the tabular NAS-Bench-201 search space, including an in-depth evaluation of NES by comparing it to the deep ensemble of the best architecture in the search space.

3. **Best practices for reporting important details**
   (a) Did you report how you tuned hyperparameters, and what time and resources this required? [Yes] We relied on default hyperparameter choices from Liu *et al.* [39] as described in Appendix B.
   (b) Did you report the time for the entire end-to-end NAS method (rather than, e.g., only for the search phase)? [Yes] See Figure 7 and Appendix B.3.
   (c) Did you report all the details of your experimental setup? [Yes]