# OpenReview forum: "Neural Ensemble Search for Uncertainty Estimation and Dataset Shift"
_NeurIPS.cc/2021/Conference — NeurIPS 2021 Poster_

### Official Review · Reviewer_28yA · 2021-07-12

**Rating:** 7
**Confidence:** 4

**Summary:**

This paper proposes a novel approach for searching ensembles of neural networks to improve the quality of uncertainty estimation and  performance improvement under the dataset shift. The main idea of the paper is to construct a pool of architectures, from which a new ensemble member is selected using a forward greedy algorithm. The paper is empirical, and all claims are strongly supported by thorough experiments.

**Ethical Concerns:**

No ethical concerns.

**Limitations And Societal Impact:**

The main limitation of this work is the computational cost of the introduced method. This, however, has been discussed by the authors. Other limitations include the concerns raised in the main review, and may only require textual changes in the manuscript.

**Main Review:**

First, this is a great paper, and it has been my pleasure to read it. I also highly appreciate the authors releasing the code.


I have only two points that would be great to discuss (or add additional results if possible) in the camera-ready version:

1. What is the effect of data augmentation? I have understood from the paper that no augmentations have been used. Why is that so? Is there an ablation study in the supplementary that I am missing?

2. It would be important to see the results on selecting an ensemble with NAS when $D_{val}^{shift}$ is not used for model selection. Since the proposed method used the $D_{val}^{shift}$, and Deep Ensembles did not, it does not seem to be a fair comparison. One can easily add a term to DEs model selection where $D_{val}^{shift}$ is used. Please discuss this in the paper.


**Time Spent Reviewing:**

4

---

> ### Author Response · Authors · 2021-08-10
> **Reply to reviewer 28yA**
>
> Thank you very much for your insightful comments and kind review! We are happy that you found this work to be a *“great paper”,* *“pleasure to read”* and that *"all claims are strongly supported by thorough experiments"*. Below is our response to the two points you recommend us to add to the paper. The new experimental results/plots are here: https://github.com/anonymous-nes/nes-neurips21
> 1. **“What is the effect of data augmentation? I have understood from the paper that no augmentations have been used. Why is that so? Is there an ablation study in the supplementary that I am missing?”** In our experiments, we didn’t use data augmentation for CIFAR-10 and CIFAR-100, but we did use standard data augmentation for Tiny ImageNet as it is a harder dataset. NES outperformed deep ensembles in both cases, and we don’t think data augmentation impacts NES negatively. In short, data augmentation can improve the individual base learners, which in turn would improve the ensemble. We described the use of data augmentations in Appendix B.3. We will explicitly discuss that data augmentation does not impact NES to improve the text clarity.
>
> 2. **“It would be important to see the results on selecting an ensemble with NAS when D_{shift}^{val} is not used for model selection. Since the proposed method used the D_{shift}^{val}, and Deep Ensembles did not, it does not seem to be a fair comparison. One can easily add a term to DEs model selection where D_{shift}^{val} is used. Please discuss this in the paper.”**: For DeepEns (RS), we actually always used D_val^shift to select the best architecture by validation performance before ensembling it, so that the deep ensemble baseline did have access to shifted validation data. We will make this clearer in the paper. Related to the use of D_val^shift, note that we included an ablation in Appendix C.2 where we don’t use D_val^shift during the pool construction stage of NES-RE, calling this variant NES-RE-0. NES-RE-0 continues to outperform deep ensemble baselines but underperforms NES-RE.
>
> Thank you again for your review! We hope to have answered your questions and will address them in the paper as well. Please let us know if you any further questions.

---

### Official Review · Reviewer_WTyi · 2021-07-16

**Rating:** 6
**Confidence:** 5

**Summary:**

This paper introduces ‘Neural Ensemble Search’, a method for (automatically) constructing ensembles of NNs which (in contrast with most iterations of deep ensembles) generally do not have the same architecture. The authors show that their method consistently outperforms baseline ensemble methods in accuracy and uncertainty calibration. They hypothesize that the success of this method is mainly due to the varied architectures leading to increased diversity of the predictive distributions of ensemble members, and they provide evidence supporting this claim.

**Main Review:**

From a practical standpoint, the method proposed offers modest gains over baselines, particularly considering the high training cost associated with producing the pool of candidate models (though this is less important than test-time cost, which is comparable to other methods, and much of the training can be parallelized).

However, the contributions of this paper largely outweigh this limitation. The relationship between diversity and performance has emerged as an important topic in understanding ensembles. This paper takes a significant step in addressing the effect of varied architectures on ensembles, which has not been previously studied in detail to my knowledge. The authors are thorough in their experiments, both to show their method outperforms baselines and to attempt to isolate the effects of architecture.

Some ideas that could potentially strengthen this submission:

* Experimental results for ensemble methods in which architecture varies, but ‘ForwardSelect’ is not used. Some examples are described in lines 198-204. Such results would complement the ablation experiments in which you use ForwardSelect over initializations for Deep Ensembles, and (one hopes) further support the claim that the variation in architecture, rather than the selection algorithm, is the main reason for the success of this method

* More analysis focusing on the architecture in detail. For example, is there a clear relationship between how different two models’ architectures are and how diverse their predictions are? I would be very interested in a t-SNE or UMAP plot, similar to that in Figure 2a, where each dot in a color represents a single mutation from the initial architecture instead of a different initialization of it. As another example, are there characteristics that tend to be shared among high performing baseline learners, and do similar architectures (differing by only a few mutations) have similar performances? Such insights could conceivably be leveraged into even better strategies for producing a pool of models and for model selection.

** Update after author response period ** Thanks to the authors for their replies. I remain positive about the work and keep my original recommendation.

**Time Spent Reviewing:**

2

---

> ### Author Response · Authors · 2021-08-10
> **Reply to reviewer WTyi**
>
> Thank you very much for your useful feedback! We’re happy to hear that you believe *“the contributions of this paper largely outweigh this limitation”*, that this work takes a *“significant step in addressing the effect of varied architectures on ensembles”* and that our experiments are *“thorough”*. Regarding your points for improvement, please see our replies below. The new experimental results/plots are here: https://github.com/anonymous-nes/nes-neurips21
>
> 1. **“Experimental results for ensemble methods in which architecture varies, but ‘ForwardSelect’ is not used. Some examples are described in lines 198-204. Such results would complement the ablation experiments in which you use ForwardSelect over initializations for Deep Ensembles, and (one hopes) further support the claim that the variation in architecture, rather than the selection algorithm, is the main reason for the success of this method”**: Note that ForwardSelect is *necessary* for NES to perform well, to ensure that the ensemble chosen doesn’t include *weak* architectures. However, our ablations show that ForwardSelect is not *sufficient* to improve performance (i.e. one also needs architectural variation), since NES outperforms deep ensembles even if they use ForwardSelect. Related to this, we have also added experimental results that compare the performance of NES for different choices of the ensemble selection algorithm. Please refer to the new Figure 1.
>
> 2. **“Is there a clear relationship between how different two models’ architectures are and how diverse their predictions are? I would be very interested in a t-SNE or UMAP plot, similar to that in Figure 2a, where each dot in a color represents a single mutation from the initial architecture instead of a different initialization of it. As another example, are there characteristics that tend to be shared among high performing baseline learners, and do similar architectures (differing by only a few mutations) have similar performances?”**: Thank you for this very interesting suggestion—we have produced the plots you requested. We sampled five random architectures from the DARTS space, and for each one, we applied a single random mutation twenty times, yielding a “family” of parent-children architectures, which are then trained on CIFAR-10. In the new Figure 4, the left plot shows t-SNE applied to the test predictions of these architectures, where each color corresponds to one “family”, of which the “star” is the parent architecture and the circles are the children architectures. The clustering demonstrates that architectures which differ by only a single mutation are similar in function space after training. The right plot shows the NLL and error achieved by these architectures. Again, similar clustering shows that architectures differing by a single mutation also perform similarly w.r.t. NLL and error. The findings of the right plot are consistent with prior work, which has shown that various search spaces exhibit “locality”, that is, mutated versions of an architecture perform similarly (Section 3.3 in [1]). However, regarding your question about high performing base learners sharing characteristics, note that the loss landscape of NAS spaces has multiple local minima [2]. Therefore, high performing architectures don’t necessarily share characteristics. Indeed, as visualized in Figure 23 in our appendix, NES-RE can end up picking diverse architectures to ensemble, although there are sometimes repetitions (the reduction cell is the same for the first and third base learners in Figure 23). We will add this new figure and discussion to the paper. Thanks again for making this suggestion!
>
> **References:** \
> [1] Ying et al., 2019: NAS-Bench-101: Towards Reproducible Neural Architecture Search \
> [2] White et al., 2021: Exploring the Loss Landscape in Neural Architecture Search
>
> Thank you for your review and the suggestions to strengthen our submission! We hope to have addressed your points for improvement, and that you will consider increasing your rating. Please let us know if you have any further questions.

---

### Official Review · Reviewer_xqD9 · 2021-07-17

**Rating:** 6
**Confidence:** 4

**Summary:**

The paper proposes neural ensemble search (NES) for adding architecture variations in forming ensembles. The main observation is that the architecture variations can be more helpful in terms of accuracy and uncertainty estimation, due to their potentially more diverse predictions, compared with constructing ensembles with a fixed architecture and based on random initializations.

**Limitations And Societal Impact:**

Yes

**Main Review:**

Improving uncertainty estimation of neural networks which the paper targets is an important problem. The proposed method builds a pool of architectures, either by random sampling or an evolutionary method, and selects an ensemble of smaller size with a greedy approach. Both the evolutionary method and greedy ensemble selection are based on previous work, without any major modification.

Using diverse models for predictive tasks is common but the work seems to be the first to systematically focus on and evaluate the effect of varying architectures in ensembles within a NAS space on improving accuracy and uncertainty estimates.

The paper is written clearly and describes the methods and choices made in the design of the method well. It performs several ablation studies to show the effect of the different parts of the method and details about the experimental setups are provided.

There are some improvements that can be made in the experiments.
For the DARTS dataset shift experiments, to isolate the effect of ensembles based on one architecture vs. multiple, DeepEns can be tested with the best architecture according to error on shifted validation data among architectures trained in the proposed NES methods.

In Figure 7, showing only the number of networks trained may not be the best metric as some methods are embarrassingly parallel. Also, why the DeepEns (RS) versions are expected to perform better than other DeepEns versions when there is no dataset shift? The ablation study is important in empirically validating the paper’s claim, so it would be good to report similar results for more than one setup and dataset and also to test DeepEns with the best architecture among the ones which NES methods select from.

The ECE results (Figure 5), and predictive disagreement (Table 2) do not conclusively demonstrate that the proposed methods are performing better in terms of uncertainty and diversity. Experiments on common OOD setups can help the claims about better uncertainty estimation.

Overall, although the methodological contributions of the paper are not very significant, the execution and the results are valuable.


--After author response--

Thanks to the authors for their replies. Including the additional results and discussion will further improve the paper and support its claims. I remain positive about the paper.



**Time Spent Reviewing:**

3

---

> ### Author Response · Authors · 2021-08-10
> **Reply to reviewer xqD9**
>
> Many thanks for your feedback and comments! We are happy you found the paper to be “*written clearly*” and are glad you agree that “*the paper targets an important problem*” and is *"the first to systematically focus on and evaluate the effect of varying architectures in ensembles within a NAS space on improving accuracy and uncertainty estimates.”* Please see our replies to your individual points for improvements below. The new experimental results are here: https://github.com/anonymous-nes/nes-neurips21
>
> 1. **“For the DARTS dataset shift experiments, to isolate the effect of ensembles based on one architecture vs. multiple, DeepEns can be tested with the best architecture according to error on shifted validation data among architectures trained in the proposed NES methods.”**: Over the DARTS space, when evaluating DeepEns (RS) on shifted data, we did indeed pick the best architecture by *shifted* validation loss *among the ones already trained by NES-RS*, and then formed a deep ensemble using it. This results in a fair comparison because the baseline has access to the same architectures as NES. It also avoids having to randomly sample, train and evaluate architectures once for each of DeepEns (RS) and NES-RS. We mentioned this in the README of our codebase, but will also add it to the paper.
>
> 2. **“In Figure 7, showing only the number of networks trained may not be the best metric as some methods are embarrassingly parallel.”**: Figure 7-left has the allowed ensemble size on the x-axis, not the number of networks evaluated. Each point on the y-axis represents the performance of the ensemble of size M that can be achieved by the different methods. Figure 7-right (the table) indeed does list the number of networks trained, but all methods shown are easily parallelized. Indeed, all RS methods (including NES-RS) can exploit an arbitrary number of parallel workers, and the RE methods (AmobeaNet and NES-RE) also enjoy perfect parallelization up to the population size. For example, for NES-RE, to fit our computational resources, we used a population size of 20, but RE methods are also known to scale to more parallel workers (using a larger population size). Finally, for the baseline DeepEns (DARTS), we agree that DARTS is not perfectly parallelizable, and while it evaluates thousands of architectures in the one-shot model, we listed the equivalent computation cost (GPU hours) of this phase in terms of full architecture evaluations; we are very open to suggestions of how to better present its compute requirements. Thanks!
>
> 3. **“Why the DeepEns (RS) versions are expected to perform better than other DeepEns versions when there is no dataset shift?”**: We found the performance of DeepEns (DARTS/AmoebaNet) to be dataset dependent; they are strong on CIFAR-10 and CIFAR-100, outperforming DeepEns (RS) significantly, however they are weaker on Tiny ImageNet. Differentiable NAS methods such as DARTS are known to have robustness issues when transferring from the one-shot model to the discrete architecture [1, 2]. In fact, on NAS-Bench-201, DARTS is known to generate degenerate architectures (see Table 5 of [3]). That’s why we compared NES to the deep ensemble of the (globally) best architecture in the search space, which was fortunately possible for NAS-Bench-201. Thanks for pointing this out, we will address this observation in the experiments section.
>
> 4. **“The ablation study is important in empirically validating the paper’s claim, so it would be good to report similar results for more than one setup…”**: Thanks for this suggestion! We’ve performed the exact same ablation study in the paper’s Figure 7 also on the CIFAR-10 dataset (since it is a different setup: smaller input dimensionality, fewer samples and fewer classes than Tiny ImageNet) now, with similar findings. NES algorithms perform amongst the best methods, with NES-RE outperforming all other methods by a clear margin as before. Please see the new Figure 6.
>
> 5. **“It would be good to test DeepEns with the best architecture among the ones which NES methods select from.”**: Please refer to the first point. DeepEns (RS) indeed consists of selecting the best (trained) architecture by validation loss amongst those that NES-RS selects from.
>
> 6. **“The ECE results (Figure 5), and predictive disagreement (Table 2) do not conclusively demonstrate that the proposed methods are performing better in terms of uncertainty and diversity.”**: We used two metrics (predictive disagreement and oracle ensemble loss) to measure diversity. Table 2 shows that for almost all datasets, NES finds more diverse ensembles wrt these metrics. Please also see Figures 15-17, which show higher diversity (lower oracle ensemble loss) consistently across datasets and ensemble sizes. Regarding ECE, we reported all our results (without nit-picking!), and NES was among the best performing methods for most dataset and shift severity configurations. There are larger improvements wrt CIFAR-10 (upto 40% better ECE) than wrt Tiny ImageNet, where anchored ensembles perform well. However, anchored ensembles perform significantly worse wrt loss (Figure 4(a)(b)-left) too.
>
> 7. **“Experiments on common OOD setups can help the claims about better uncertainty estimation.”**: In our experiments, our primary focus was *predictive* uncertainty estimation, measured using NLL (which is uncertainty-aware, unlike classification error) and ECE. However, we also did briefly explore OOD detection in Appendix C.1.2 and Figure 12 by measuring the entropy of the predicted class label probability distributions for OOD inputs. While NES and deep ensembles performed similarly for OOD inputs from SVHN, NES was better at detecting OOD inputs from Gaussian noise than deep ensembles.
>
> **References:** \
> [1] Yu et al., 2019: Evaluating the Search Phase of Neural Architecture Search \
> [2] Zela et al., 2020: NAS-Bench-1Shot1: Benchmarking and Dissecting One-shot Neural Architecture Search \
> [3] Dong & Yang, 2020: NAS-Bench-201: Extending the Scope of Reproducible Neural Architecture Search
>
> Thank you again! We hope we have addressed your concerns and that you will consider increasing your rating. Please let us know if you have any further questions.

---

### Official Review · Reviewer_dkBq · 2021-07-17

**Rating:** 6
**Confidence:** 4

**Summary:**

This paper proposes neural network ensembles with varying architectures, arguing these will be produce better uncertainty estimates, and considers various straightforward techniques to aggregate (ie. ensemble) the predictions from a pool of independently trained heterogeneous neural network architectures. The aggregation algorithm in this work is simply Ensemble Selection from Caruana et al., and random search / evolution methods are considered for automatically constructing the neural architectures in the base learner pool, rather than requiring these are predefined.


**Ethical Concerns:**

I don't have any remaining omitted ethical concerns regarding this paper.

**Limitations And Societal Impact:**

I believe the authors have adequately addressed societal impact of their work.
However a key limitation of the proposed ensembling is that inference latency will be slower for deploying a neural ensemble vs a single network, and may not be suitable for real-time applications. It would be insightful if the authors could compare their ensemble against a single large network where both models are restricted to a total number of NN parameters (summed across all ensemble members in the former case) or restricted to the same inference latency.


**Main Review:**

Two key questions are considered in this paper:

Q1) How to generate a pool of neural networks to ensemble?
Q2) How to aggregate the predictions from this NN pool into a single prediction?

There is little theory or conceptual justification for the authors' choices to address each question, so the empirical comparison must be sufficiently convicing to justfy the contribution of this paper. However, I do not find the comparisons sufficently thorough in the current paper.

Furthermore, the novelty of the proposed methods is limited (which would not be a problem if the proposed straightforward methods showed clear superiority over existing state of the art, but this is not clearly demonstrated in my opinion). A standard approach in eg AutoML systems is to do hyperparameter optimization (HPO) followed by ensembling of the models corresponding to best hyperparameter candidates. This work essentially considers the same framework, except the hyperparameter search space now includes neural architectures and rather than Bayesian optimization HPO algorithms, simple NAS algorithms are considered for searching through this space.

Regarding Q2: it is unclear what is specific to the base learners being NN models, and it seems like any arbitrary ensembling algorithm could apply. The authors here advocate for Ensemble Selection (ES) from Caruana et al., which is certainly a reasonable choice, but it would be good to see more comparisons against alternative standard ensembling algorithms. The alternative ensembling algorithms considered by the authors as baselines vs. ES are too limited and non-adaptive unlike many ensembling algorithms which exist today (and where are the empirical results that back up the claim: " these three performed comparatively or worse
204 than our choice ForwardSelect"?). Here are some straightforward alternatives off the top of my head:

A) Train a linear model or neural network to combine the predictions. Ie. stacking, unclear why this left for future work as it is simple to try out, see for example:

AutoGluon-Tabular: Robust and Accurate AutoML for Structured Data.
Erickson et al. 2020.
https://arxiv.org/abs/2003.06505

where this has been shown to be highly performant method for ensembling heterogeneous base learners. Also blending is another standard alternative to stacking. AutoGluon also appears to use Bayesian optimization to tune architectural hyperparameters of NN base learners (contrary to what the authors write in the "AutoML and ensembles of varying architectures" section).

B) Various voting ensemble schemes that have been previously proposed, eg:
https://scikit-learn.org/stable/modules/generated/sklearn.ensemble.VotingClassifier.html

Furthermore, it seems promising to exploit the fact that the base learners are all NN models to consider more interesting differentiable aggregators such that more interesting overall strategies can be considered like those of Webb et al. [49]. For example, rather than purely ensembling based on validation score, perhaps the aggregator should also consider some properties of the architectures being ensembled which might lead to better OOD uncertainty estimation, something that is not easily achieved solely thru validation-score ensemble construction.

The authors should also justify why they aggregate predicted probabilities instead of say logits, and why a simple average of these is considered rather than say a weighted combination. For example: in the baseline "Select the top M networks by validation performance", will performance improve if the resulting predictions of the networks weighted by their validation performance in the aggregator (as in Bayesian model averaging)?


Regarding Q1:

- As a baseline, why not use standard NAS algorithms (eg. ENAS, ProxylessNAS) to generate the pool? This could still be followed by ensemble-selection just as applied to pools considered in this paper. I suppose one issue is the weight-sharing scheme in standard algorithms like ENAS but it would still be insightful to see how ensembles of these weight-shared candidates perform.

- Does the pool improve if bootstrapping is used to train each NN base learner on a different subsampled dataset? This would presumably increase diversity.

- For efficiently generating more pool candidates, you could also consider ensemble selection applied to all NN checkpoints (eg. every few epochs) rather than only fully trained NN base learners. Such checkpoint-ensembles are known to be superior to individual NN models.


- Other comparisons the paper should consider:

While briefly cited as a related work, Webb et al. 2019 [49] needs to be discussed in more detail, and conceptually constrasted against the proposed ideas in this paper. You only cite [49] as evidence that joint training of ensemble is worse than indpeendent training of base learners, but [49] does a much deeper analysis and finds some amount of joint training (ie. end-to-end learning) combined with independent training can lead to the best results.

Appendix C.4 should also compare against another version of HyperEns, where similar architectural design search spaces as considered in NES are considered as hyperparameters to be ensembled in HyperEns. This is a trivial modification of HyperEns and is a highly relevant baseline to compare against (probably should even appear in the main paper).

- How were hyperparameters chosen in this work?

- When it comes to epistemic uncertainty (ie. model uncertainty), the typical assumption is that model family (ie. hypothesis class) is correctly specified and we just don't have enough data to properly identify the right model from the class (this is sometimes called the "M-closed" assumption in Statistics). A major advantage of ensembling heterogeneous models as done here would be the ability to provide better uncertainty estimates under model misspecification, which is always present in practice (especially with neural network models). I think the authors could mention this connection and cite some related literature in this area. I'm not sure Proposition 3.1 is the best way to motivate having more diverse ensembles, as there are other standard arguments in favor of diversity presented in the ensembling literature (eg. statistical independence of base learner errors).


- A remaining concern I have is the following: This paper seems to train networks on D_train and the construct ensemble based on D_val, and finally score the ensemble on D_test. However an alternative standard procedure is to refit all networks on D_train+D_val before evaluating on D_test. I suspect DeepEnsembles would fare better under this alternative than Ensemble-Selection whose outperformance is more contingent on the base learners behaving similarly when fit to just D_train vs D_train+D_val. It would be good to address this point.


- There is also much related work in which ensembles of heterogeneous NN architectures have been found useful that could be referenced. Here's just a couple papers I found thru google:

Ensembles of Networks Produced from Neural Architecture Search.
Herron et al. 2020.
https://link.springer.com/chapter/10.1007/978-3-030-59851-8_14

Ensembling neural networks: Many could be better than all.
Zhou et al. 2002.
https://www.sciencedirect.com/science/article/pii/S000437020200190X?via%3Dihub

Label Denoising with Large Ensembles of Heterogeneous Neural Networks.
Ostyakov et al. 2019.
https://arxiv.org/abs/1809.04403

Antibody Complementarity Determining Region Design Using High-Capacity Machine Learning.
Liu et al. 2020.
https://academic.oup.com/bioinformatics/article/36/7/2126/5645171

Solo or Ensemble? Choosing a CNN Architecture for Melanoma Classification.
Perez et al. 2019.
https://openaccess.thecvf.com/content_CVPRW_2019/html/ISIC/Perez_Solo_or_Ensemble_Choosing_a_CNN_Architecture_for_Melanoma_Classification_CVPRW_2019_paper.html

An ensemble of neural networks for weather forecasting.
Maqsood et al. 2004.
https://link.springer.com/article/10.1007/s00521-004-0413-4

NADS: Neural Architecture Distribution Search for Uncertainty Awareness.
Ardywibowo et al. 2020.
https://arxiv.org/abs/2006.06646


### Update after reading author response to my original review ###

I am glad to see the authors' additional experiments that addressed my major concerns and have increased my score. I encourage the authors to include all of the points in their response in the revised paper (can squeeze these into appendix as necessary), as I think they will be useful to future readers. And I encourage the authors to tone down any claims of being the first work to ensemble diverse architectures and instead highlight the novelty of this work in terms of discovering the utility of architecture-ensembles for better uncertainty estimation and their new ensemble-aware architecture search algorithm.

Regarding: "perhaps the aggregator should also consider some properties of the architectures being ensembled which might lead to better OOD uncertainty estimation, something that is not easily achieved solely thru validation-score ensemble construction"

One idea could be look at the correlation/variance in the individual ensemble-members predictions as an auxiliary diversity objective (or take a weighted combination of validation-score objective and diversity objective). Some examples are considered in these works:

Brown, G. 2004. Diversity in neural network ensembles. Ph.D. Dissertation, University of Birmingham.

Jain et al. 2020. Maximizing Overall Diversity for Improved Uncertainty Estimates in Deep Ensembles. https://arxiv.org/abs/1906.07380

**Time Spent Reviewing:**

7

---

> ### Author Response · Authors · 2021-08-10
> **Reply to reviewer dkBq (part 1)**
>
> Thank you very much for your detailed review! Below we respond to your individual points. The new experimental results/plots are here: https://github.com/anonymous-nes/nes-neurips21
> 1. **“Novelty of the proposed methods is limited… A standard approach in e.g. AutoML systems is to do hyperparameter optimization (HPO) followed by ensembling of the models corresponding to best hyperparameter candidates. This work essentially considers the same framework, except the hyperparameter search space now includes neural architectures and rather than Bayesian optimization HPO algorithms, simple NAS algorithms are considered for searching through this space.”** Indeed, automatically constructing ensembles using ensemble selection is common in the context of AutoML. We hoped to convey that in lines 90-95. However, our work’s contribution is to (1) consider the impact of ensembles with varying architectures on *uncertainty estimation and robustness* and (2) search for such architectures over *complex, SOTA* architecture search spaces. We believe the former is especially important, because deep ensembles are a SOTA and increasingly popular method for uncertainty estimation. To our knowledge, prior work in AutoML focused only on predictive performance, and methods that considered architectural variation were limited to only varying depth/width or fully-connected networks.
>
> 2. **Comparing different ensemble selection algorithms, stacking, weighted averaging and Bayesian model averaging: "it would be good to see more comparisons against alternative standard ensembling algorithms… ”, “where are the empirical results that back up the claim [in line 204]?”, “Train a linear model or neural network to combine the predictions. Ie. stacking”, “will performance improve if the resulting predictions of the networks weighted by their validation performance in the aggregator (as in Bayesian model averaging)?”**: Thanks for this suggestion; we agree that a comparison of ensemble selection algorithms (ESAs) for NES is a useful addition to the paper! Therefore, we’ve added results (in the new Figure 1) comparing the performance of various ESAs for stage 2 of the NES algorithms. To support our claim in line 204, this includes the three options—ForwardSelect *with* replacement, “top M” and a “quick and greedy” algorithm—described in line 204, which are shown here to underperform or be at par with ForwardSelect. As you suggested, we also implemented weighted/unweighted stacking and ForwardSelect weighted by validation likelihood/classification accuracy (i.e. Bayesian model averaging). The results are shown in new Figure 1 along with a description of all the ESAs that we experimented with. Takeaways from these results: (1) ForwardSelect tends to outperform or be at par with all the other ESAs. “Top M”, which simply selects the top M base learners by their individual validation performances, unsurprisingly performs worst, as it is ensemble-unaware. The “quick and greedy” ESA also performs poorly, likely again due to its preference for strong base learners. ForwardSelect *with* replacement ends up selecting precisely the same base learners *without* replacement. (2) Stacking tends to perform competitively but still worse than ForwardSelect. Prior work (Section 3.2 of [1]) has also compared various ESAs (in a different context) and similarly found stacking to underperform compared to ForwardSelect. Whether the stacking is weighted or not has a very minor impact on its performance. (3) Bayesian model averaging also has a very minor impact (almost invisible in the plot), because the likelihoods and accuracies of individual base learners are very similar (a consequence of multi-modal loss landscapes in neural networks with different models achieving similar losses) hence the BMA weights are close to uniform. We will add these results and discussion to the paper; thank you!
>
> 3. **“AutoGluon also appears to use Bayesian optimization to tune architectural hyperparameters of NN base learners”**: Thanks for mentioning AutoGluon, we will add this to our related work section! Please note that the default version of AutoGluon (https://arxiv.org/abs/2003.06505) does not do hyperparameter tuning: “While AutoGluon does support various hyperparameter optimization strategies, we do not utilize its hyperparameter_tune = True option in this work.” From previous experience and correspondence with the authors of AutoGluon, we found that turning hyperparameter tuning on significantly worsens its performance.
>
> 4. **“... rather than purely ensembling based on validation score, perhaps the aggregator should also consider some properties of the architectures being ensembled which might lead to better OOD uncertainty estimation, something that is not easily achieved solely thru validation-score ensemble construction.”**: Thanks for this interesting idea—we are keen to explore this! Do you have suggestions for other aggregation techniques incorporating such “meta”-information? It could perhaps be useful to add some explicit architectural diversity regularization? We are open to suggestions and can quickly add new results (since we only need to run any new ensemble selection algorithms on already-trained pools, which is fast).
>
> 5. **“The authors should also justify why they aggregate predicted probabilities instead of say logits, and why a simple average of these is considered rather than say a weighted combination.”**: Regarding averaging logits vs. probabilities, we found that ensembles averaging logits perform worse than ensembles averaging probabilities, particularly wrt uncertainty estimation. See the new Figure 3, which shows that NLL and ECE of logit-averaged ensembles is significantly worse, and classification error is mostly unchanged. Averaging probabilities also has a useful interpretation as a uniform mixture model, as discussed in Section 2.4 of [2], and it is also used in deep ensembles. Regarding simple vs. weighted averaging: since our aim was to isolate and understand the improvement in performance due to varying architectures when compared with SOTA deep ensembles, we used simple averaging since deep ensembles also do that. This also has the benefit of being straightforward yet highly performant in practice. However, upon your suggestion, we also tried weighted stacking and two types of Bayesian model averaging, please see point 2 above. We will add this discussion and figure to the paper. Thanks very much!
>
> 6. **“As a baseline, why not use standard NAS algorithms (eg. ENAS, ProxylessNAS) to generate the pool? This could still be followed by ensemble-selection just as applied to pools considered in this paper. I suppose one issue is the weight-sharing scheme in standard algorithms like ENAS but it would still be insightful to see how ensembles of these weight-shared candidates perform.”**: Exhaustively evaluating NAS algorithms for generating the pool is outside the scope of our work, because our aim is to show that architectural variation is beneficial in its impact on uncertainty estimation and robustness compared to deep ensembles. To do that, as a first step, we tried random search, which already outperforms deep ensembles in our experiments, then we built on that using a modification of regularized evolution. (Note we didn’t use standard regularized evolution; NES-RE steers the evolution of the population in an ensemble-aware way.) Nonetheless, various NAS algorithms can indeed be used to generate the pool, and it would be interesting to explore that. We are particularly excited about using differentiable NAS algorithms to reduce the cost of NES as discussed in our conclusion. Nonetheless, we conducted an additional experiment, which generates the pool using the weight sharing schemes proposed by Random Search with Weight Sharing [3] and DARTS [4]. Specifically, we trained one-shot weight-sharing models using each of these two algorithms, then we sampled architectures from the weight-shared models uniformly at random to build the pool. Next we ran ForwardSelect as usual to select ensembles from these pools. Finally, we retrained the selected base learners from scratch using the same training pipeline as NES and deep ensemble baselines. The results of these methods are referred to as NES-RS-WS and NES-RS-DARTS respectively in the new Figure 2. As shown, the performance is worse than NES-RS and NES-RE, even though the computation cost is reduced significantly. Precisely as you pointed out, this is likely due to weight-sharing aspect. Specifically, there can be low correlation between the performance of the architectures when evaluated using the shared weights and the performance when re-trained in isolation. Prior work has shown this is caused by weight interference and coadaptation which occurs during the weight-shared model’s training [5, 6]. We still believe this is an interesting direction to pursue in order to reduce the compute cost of NES, but it requires more development, which is outside the scope of our current work.
>
> 7. **“Does the pool improve if bootstrapping is used to train each NN base learner on a different subsampled dataset? This would presumably increase diversity.”** Prior work has found that bootstrapping does *not* work as well as training individual deep networks. This point was explored in Section 2.4 of [2], which argued that bootstrapping reduces the number of unique data points each base learner is trained on, which worsens performance. Subsequent work [7] dedicatedly studied why bootstrapping doesn’t work in neural networks, concluding “We investigate a common hypothesis for bootstrap’s weak performance—percentage of unique points in the subsampled dataset. We find that even when adjusting for it, bootstrap ensembles of deep neural networks yield no benefit over simpler baselines.” Therefore, we did not use bootstrapping.
>
> [continued below]

---

> > ### Author Response · Authors · 2021-08-10
> > **Reply to reviewer dkBq (part 2)**
> >
> > 8. **“For efficiently generating more pool candidates, you could also consider ensemble selection applied to all NN checkpoints (eg. every few epochs) rather than only fully trained NN base learners. Such checkpoint-ensembles are known to be superior to individual NN models.”**: Thank you for this suggestion. While this would desiredly reduce training cost, recent work [8] compared deep ensembles with snapshot ensembles (i.e. checkpoint-ensembles), finding that “many sophisticated ensembling techniques are equivalent to an ensemble of only few independently trained networks in terms of test performance”. Please see Fig. 3 in [8], which shows that deep ensembles outperform snapshot ensembles on each of CIFAR-10, CIFAR-100 and ImageNet for a fixed ensemble size.
> > 9. **“You only cite Webb et al., 2019 [49] as evidence that joint training of ensemble is worse than independent training of base learners, but [49] does a much deeper analysis and finds some amount of joint training (ie. end-to-end learning) combined with independent training can lead to the best results.”**: Webb et al., 2019 [9] indeed did a thorough analysis exploring joint training vs. independent training, However, they found “These results indicate there is little to no benefit in test error when E2E [end-to-end, i.e. joint] ensemble training SOTA deep neural networks.” (Section 4, page 7). Some amount of joint training was only observed to be beneficial for MLPs and small/moderate CNNs in their work, but not deep CNNs. Moreover, joint training is not embarrassingly parallel and has a large memory cost (Webb et al., 2019 used ensemble size 4 for their larger networks due to memory constraints). We therefore did not explore joint training, but we will include this discussion in our paper for completeness.
> > 10. **“Appendix C.4 should also compare against another version of HyperEns, where similar architectural design search spaces as considered in NES are considered as hyperparameters to be ensembled in HyperEns.”**: As discussed in our related work, hyperparameter ensembles is concurrent work (appeared on arXiv after ours). Their work showed that varying non-architectural hyperparameters, such as weight decay and dropout probability, is beneficial, whereas our work shows that varying architecture is beneficial. The question of which hyperparameters to vary is very interesting but outside the scope of both works. Our results in the appendix are a preliminary step in that direction, but a detailed analysis is left for future work, as our focus was on architectural variation.
> > 11. **“How were hyperparameters chosen in this work?”**: All hyperparameter choices for training base learners are in Appendix B (specifically B.3 and lines 669-687), chosen in most cases to be the default settings used by DARTS (unless otherwise specified) or NAS-Bench-201. Other hyperparameters such as ensemble size M and training budget K were varied, with performance shown as a function of them. Is there any particular hyperparameter you are curious about?
> > 12. **“A major advantage of ensembling heterogeneous models as done here would be the ability to provide better uncertainty estimates under model misspecification, which is always present in practice (especially with neural network models). I think the authors could mention this connection and cite some related literature in this area.”** Thanks for this great suggestion! We fully agree; model mis-/under-specification can exist not just with respect to parameters of a fixed architecture but the choice of architecture itself too. We will add a paragraph to discuss this in the introduction and cite related literature, as it helps motivate/conceptualize NES.
> > 13. **“However an alternative standard procedure is to refit all networks on D_train+D_val before evaluating on D_test. I suspect DeepEnsembles would fare better under this alternative…”** We trained the ensembles constructed by NES and the best performing deep ensemble baseline, DeepEns (DARTS), on D_train + D_val, finding that both ensembles improve since the base learners improve due to more training data. The results are in the new Figure 5. Note that, as shown in Appendix C.5, the performance of ensemble selection (which uses D_val) is relatively insensitive to the size of D_val. Therefore, one way to bypass the additional cost of having to retrain the ensembles on D_train + D_val is to simply pick a very small D_val, such that the performance of the models when trained on D_train + D_val is approximately the same as when trained on D_train.
> > 14. **Regarding other related work**: Thank you for the related work suggestions; we will add them to the paper. Most of these works focus on applications and utilize heterogeneous ensembles for boosting performance, though without automatic architecture selection.
> > 15. **“It would be insightful if the authors could compare their ensemble against a single large network where both models are restricted to a total number of NN parameters (summed across all ensemble members in the former case) or restricted to the same inference latency.”**: The question of whether to ensemble or use a larger network of the same size as the ensemble is intriguing but not specific to our work. However, recent work [10] studied this and concluded: “Our important practical finding is that one large network may perform worse than an ensemble of several medium-size networks with the same total number of parameters (we call this ensemble a memory split).”
> >
> > **References**: \
> > [1] Feurer et al., 2019: Efficient and Robust Automated Machine Learning \
> > [2] Lakshminarayanan et al., 2017: Simple and Scalable Predictive Uncertainty Estimation using Deep Ensembles \
> > [3] Li & Talwalkar, 2019: Random Search and Reproducibility for Neural Architecture Search \
> > [4] Liu et al., 2019: DARTS: Differentiable Architecture Search \
> > [5] Yu et al., 2019: Evaluating the Search Phase of Neural Architecture Search \
> > [6] Zela et al., 2020: NAS-Bench-1Shot1: Benchmarking and Dissecting One-shot Neural Architecture Search \
> > [7] Nixon et al., 2020: Why Aren’t Bootstrapped Neural Networks Better? \
> > [8] Ashukha et al., 2020: Pitfalls of In-Domain Uncertainty Estimation and Ensembling in Deep Learning \
> > [9] Webb et al., 2019: To Ensemble or Not Ensemble: When does End-To-End Training Fail? \
> > [10] Lobacheva et al., 2021: On Power Laws in Deep Ensembles
> >
> >
> > Thank you again for your effort and detailed review! We hope we have addressed your concerns and that you will consider increasing your rating. Please let us know if you have any further questions.

---

> > > ### Author Response · Authors · 2021-08-28
> > > **Happy to have addressed your major concerns. Response to final point about ensemble selection.**
> > >
> > > We are happy that we have addressed your major concerns and appreciate that you increased your score! We will add all the points discussed to the paper.
> > >
> > > We respond to your final point regarding ways to incorporate diversity during ensemble selection from the pool; thank you for the references and useful suggestions. We added a diversity term in the optimization objective as you suggested. Please note that the work of Jain et al., 2020 focuses only on regression settings, where the network output parameterizes the mean and variance of a Gaussian distribution, which they then use to define diversity and optimize for it. Unfortunately, this is not directly applicable in classification settings such as ours (they only have experiments for regression as well). However, inspired by Brown’s work, we defined a different diversity metric which encourages each base learner’s predicted class probabilities to be far from the predicted class probabilities of the ensemble in l_2 distance. We optimize this objective using ForwardSelect. This introduces a new hyperparameter: the strength of the diversity metric in the optimization objective. The results for different choices of diversity strength are shown in the updated Figure 1 on the GitHub link (https://github.com/anonymous-nes/nes-neurips21). In summary, if appropriately tuned, ForwardSelect with diversity performs slightly better than usual ForwardSelect for NES-RS, though for NES-RE, the diversity term seems to harm performance. This suggests that there may be settings where inducing diversity in this way during ensemble selection can be beneficial, therefore we will ensure to include this in the paper with a discussion. Thanks again for your engagement in helping improve our work!

---

### Author Response · Authors · 2021-08-10
**Reply to all reviewers and area chair**

Dear reviewers and area chair,

We have now addressed the suggestions and concerns of the reviewers. Thank you very much to all the reviewers for their insightful feedback, as we believe it has much improved the paper. The changes we will make to the paper include incorporating new discussions and adding new experimental results that were suggested by the reviewers. The new figures are available here: https://github.com/anonymous-nes/nes-neurips21. Below is a summarized list of these changes:

1. **Figure 1: Comparison of ensemble selection algorithms, including ForwardSelect with/without replacement, top M, quick and greedy, weight/unweighted stacking, Bayesian model averaging (reviewers dkBq, WTyi).** ForwardSelect is consistently better than or at par with other methods. Weighted averaging tends to have very little impact on performance.
2. **Figure 2: Generating the architecture pool using weight-sharing NAS algorithms (reviewer dkBq).** We generate the pool for NES using weight-sharing. This underperforms NES-RS and NES-RE, likely due to the low correlation between the performance of the architectures when evaluated using the shared weights and the performance when re-trained from scratch.
3. **Figure 3: Averaging logits vs. averaging probabilities in an ensemble (reviewer dkBq).** Plots demonstrate that averaging probabilities is strictly better w.r.t. both NLL and ECE compared to averaging logits, while classification error remains broadly the same.
4. **Figure 4: Analysis of architecture mutations in function space and by performance (reviewer WTyi).** We empirically analyze mutated versions of an initial architecture in function space using t-SNE and by test performance. We find clear clustering in both cases, suggesting that mutations of an architecture are similar in function space and also achieve similar losses and errors.
5. **Figure 5: Training architectures on D_train + D_val (reviewer dkBq)**: We trained ensembles constructed by NES and DeepEns (DARTS) on D_train + D_val finding that both ensembles improve since the base learners improve due to more training data.
6. **Figure 6: Ablation study comparing NES with deep ensembles + ensemble selection on a second dataset, CIFAR-10 (reviewer xqD9)**: Results are similar to those with Tiny ImageNet, in that NES algorithms are still amongst the best performing methods, with NES-RE outperforming all other methods by a clear margin.
7. We will also make text changes as detailed in our individual reviewer responses.

---

### Decision · Program_Chairs · 2021-09-27

**Decision:**

Accept (Poster)

**Comment:**

This paper extends prior work on deep ensembles by introducing methods for constructing ensembles of varying architectures instead of relying on multiple random nationalizations of the same architecture. The methods introduced in the paper represent a novel combination of existing approaches applied specifically to developing architecturally diverse ensembles to improve uncertainty estimation. The writing is clear and the methods are technically correct. The authors present extensive experiments showing that the proposed method outperforms prior approaches. The reviewers had many questions and suggestions for the authors in their initial reviews. Following the discussion, the reviewers were in agreement that their primary questions had been adequately addressed and that the paper should be accepted. The authors need to be sure to include all of the discussed updates in the final version of the paper.